# FROM SPATIAL TO ACTIONS: GROUNDING VISION-LANGUAGE-ACTION MODEL IN SPATIAL FOUNDATION PRIORS

**Zhengshen Zhang**[1,2,*], **Hao Li**[3], **Yalun Dai**[3], **Zhengbang Zhu**[1], **Lei Zhou**[2], **Chenchen Liu**[2], **Dong Wang**[1], **Francis E. H. Tay**[2], **Sijin Chen**[1], **Ziwei Liu**[3], **Yuxiao Liu**[1,†,‡], **Xinghang Li**[4,†], **Pan Zhou**[5,†]

[1]ByteDance Seed, [2]National University of Singapore, [3]Nanyang Technological University, [4]Tsinghua University, [5]Singapore Management University

zhengshen_zhang@u.nus.edu    liuyuxiao.876@bytedance.com
lixingha23@mails.tsinghua.edu.cn    panzhou@smu.edu.sg

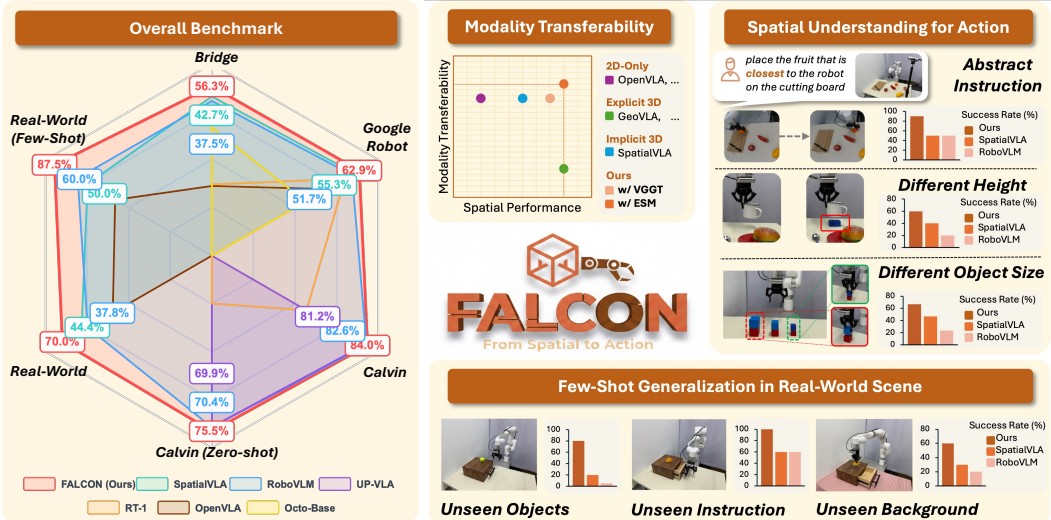

Figure 1: We propose **FALCON**, a vision-language-action model that achieves robust 3D spatial understanding by effectively integrating spatially rich tokens and semantic features. FALCON demonstrates notable modality transferability by performing robustly with both RGB-only and multi-modal inputs, superior spatial understanding in tasks involving unseen object sizes, heights and abstract spatial instructions, and strong few-shot generalizability in real-world scenes. The model achieves state-of-the-art performance across a diverse range of benchmark evaluations.

## ABSTRACT

Existing vision-language-action (VLA) models act in 3D real-world but are typically built on 2D encoders, leaving a spatial reasoning gap that limits generalization and adaptability. Recent 3D integration techniques for VLAs either require specialized sensors and transfer poorly across modalities, or inject weak cues that lack geometry and degrade vision-language alignment. In this work, we introduce **FALCON (From Spatial to Action)**, a novel paradigm that injects rich 3D spatial tokens into the action head. FALCON leverages spatial foundation models to deliver strong geometric priors from RGB alone, and includes an *Embodied Spatial Model* that can optionally fuse depth, or pose for higher fidelity when available, without retraining or architectural changes. To preserve language reasoning, spatial tokens are consumed by a *Spatial-Enhanced Action Head* rather than being concatenated into the vision-language backbone. These designs enable FALCON to address limitations in spatial representation, modality transferability, and alignment. In comprehensive evaluations across three simulation benchmarks and eleven real-world tasks, our proposed FALCON achieves state-of-the-art performance, consistently surpasses competitive baselines, and remains robust under clutter, spatial-prompt conditioning, and variations in object scale and height. Project page: https://falcon-vla.github.io/

*Works done during internship at ByteDance Seed. †Corresponding authors. ‡Project lead.

# 1 INTRODUCTION

Recent advances in vision-language-action models (VLAs) have significantly advanced the pursuit of generalist robotics, enabling robots to interpret natural language instructions and execute intricate action sequences (Brohan et al., 2023; Kim et al., 2024; Black et al., 2024; Li et al., 2024a; Zhang et al., 2024; Intelligence et al., 2025; Liu et al., 2025b). Most advanced VLAs are built on 2D foundation models like Vision Language Models (VLMs) (Liu et al., 2025a; Shao et al., 2025; Alayrac et al., 2022; Radford et al., 2021; Peng et al., 2023; Liu et al., 2023), aligning 2D images with text and leveraging the strong language understanding of the Large Language Models (LLMs) for information processing. This design provides strong semantic understanding and supports manipulation tasks conditioned on language and camera inputs.

However, while VLMs operate purely in the 2D domain, VLAs must interact with the 3D physical world. This discrepancy results in a critical gap: current VLAs lack reliable 3D spatial understanding, leading to persistent challenges in generalization and adaptability. Specifically, the absence of explicit 3D awareness causes VLAs to struggle in scenarios that require reasoning about geometry, depth, or spatial relations. *First*, they exhibit limited generalization, failing to transfer robustly to novel scenes, backgrounds, or object variations (Ze et al., 2025). *Second*, they lack adaptability to environmental changes, such as height variations or object scale differences (Li et al., 2025). These limitations now form a major bottleneck in developing reliable generalist robot policies.

To address this gap, recent works incorporate 3D information into VLAs (Li et al., 2025; Sun et al., 2025; Zhen et al., 2024), most often by providing explicit 3D inputs (e.g., point clouds or depth maps). While effective under ideal conditions, these methods suffer from low modality transferability—**the ability to function and improve under different input modalities (RGB-only, RGB-D, point clouds, camera pose) without retraining or collapsing.** This stems from two fundamental issues. First, acquiring high-quality 3D inputs requires specialized sensors that are expensive and difficult to deploy in practice. Second, many large-scale manipulation datasets (e.g., Open X-Embodiment dataset (O'Neill et al., 2024)) lack aligned 3D annotations, limiting scalability. As a result, such methods are tied to specific input modalities and break down when those inputs are unavailable.

An alternative direction introduces weak 3D cues, e.g., pseudo-depth estimates (like ZoeDepth (Bhat et al., 2023)) or learnable spatial embeddings (Qu et al., 2025). However, these approaches face three fundamental limitations. **(1) Limited spatial representation.** The learnable spatial embeddings provide only weak geometric signals within the high-dimensional space of LLMs, failing to capture robust 3D priors necessary for tasks like reasoning about height differences or object sizes in grasping. **(2) Lack of modality transferability.** While encoding some 3D cues, they cannot effectively exploit higher-quality 3D inputs when available. **(3) Challenges in alignment.** Concatenating spatial embeddings with text tokens risks disrupting the original vision–language alignment. The scarcity of 3D data makes it difficult to properly align modalities, causing embedding drift that degrades zero-shot generalization, especially in tasks requiring high-level reasoning like spatial prompts (Qu et al., 2025).

**Contribution.** We propose **FALCON** (From Spatial to Action), a novel paradigm that integrates richer and more representative 3D spatial tokens into VLAs through an improved injection scheme.

To solve limitation (1) i.e., weak 3D spatial representation, we leverage insights from spatial foundation models that encode scenes into token sequences for holistic 3D reconstruction. FALCON adopts broader and richer tokens from these foundation models, and delivers comprehensive spatial information from RGB signals alone, improving robustness when explicit 3D inputs are absent.

For limitation (2) of poor modality transferability, we introduce an Embodied Spatial Model that can optionally integrate extra 3D modalities (e.g., depth, poses). Unlike prior methods that require specific 3D inputs (Li et al., 2025; Sun et al., 2025; Zhen et al., 2024), our design is flexible: when RGB-D cameras or calibrated scenes are available, the model gains additional accuracy; when not, it still performs effectively with RGB-only input. This optional pathway substantially improves transferability by flexibly utilizing 3D signals from sensors without requiring architectural changes.

To overcome limitation (3) of alignment challenges, we draw inspiration from the brain's division of labor. The VLM (cerebrum) handles high-level reasoning and semantics, while the action head (cerebellum) manages fine-grained motor control and sensorimotor integration (Rochefort et al.,

2011; Figure, 2024). Motivated by this, we design a Spatial-Enhanced Action Head that directly incorporates spatial tokens into action decisions, which is a more natural fit since precise control depends on detailed spatial cues. This departs from prior approaches that forcibly align spatial and text tokens within VLMs (Fan et al., 2025; Wu et al., 2025).

In this way, FALCON provides (i) robust spatial reasoning, (ii) strong modality transferability, and (iii) principled integration of 3D priors into VLAs. As shown in Fig. 1, extensive experiments across three simulation benchmarks and 11 real-world tasks which including cluttered-scene manipulation, few-shot adaptation, and spatial capability evaluation, demonstrate that FALCON consistently outperforms existing baselines, achieving state-of-the-art performance with strong robustness and generalization (e.g., spatial-prompt conditioning, height-aware manipulation, object-scale variation).

## 2 METHODOLOGY

In this section, we first formulate the problem of task-oriented, language-guided robot control (Sec. 2.1). We then introduce the overall architecture (Sec. 2.2), training objective (Sec. 2.3) of FALCON, and detail its two key components: the Embodied Spatial Model (ESM) for 3D geometric representation (Sec. 2.4), and the Spatial-Enhanced Action Head for multimodal fusion and action generation (Sec. 2.5).

### 2.1 PROBLEM DEFINITION

We study the problem of *task-oriented robot control*, where a robot must interpret visual observations $O_t = \{I_t^1, \ldots, I_t^n\}$ at time step $t$ and a natural language instruction $L$ to generate an action sequence $A_t = [a_t, \ldots, a_{t+C-1}]$ through a mapping function $\mathcal{F}(\cdot)$. Each action $a_i$ is a 7D vector encoding the 6-DoF gripper pose (e.g., Euler angles) and a binary open/close state, with $C$ denoting the action horizon. Our focus is on table-top manipulation with a robot arm, using inputs from a static side camera $I_t^{3rd}$ (global scene context), a wrist-mounted camera $I_t^{hand}$ (fine-grained object details), or both. Optional depth maps $D_t \in \mathbb{R}^{H \times W}$ and camera poses $P \in \mathbb{R}^7$ can further enhance spatial grounding when available but are not strictly required, ensuring robustness across different sensing setups. Formally,

$$A_t = \mathcal{F}(O_t, L, D_t, P), \quad \text{where } D_t \text{ and } P \text{ are optional.} \tag{1}$$

This setting spans diverse applications from service robots following language commands to industrial manipulators performing instruction-driven assembly, where robust performance in unstructured environments requires tight integration of semantic understanding and geometric reasoning. To this end, we propose **FALCON**, a generalist robot policy that overcomes limitations of prior VLAs by integrating rich geometric priors from spatial foundation models while flexibly leveraging optional 3D modalities. The result is a spatially enhanced VLA that unifies semantic reasoning with fine-grained geometric grounding for robust manipulation across diverse conditions.

### 2.2 OVERALL ARCHITECTURE

As illustrated in Fig. 2, FALCON is an end-to-end VLA consists of three core components: (1) a 2D VLM for multimodal semantic representation, (2) an ESM for extracting 3D structural features, and (3) a Spatial-Enhanced Action Head that combines both streams to generate precise robot actions.

At each timestep $t$, given image observations $O_t$ and a language instruction $L$, FALCON employs a pre-trained 2D VLM (e.g., Kosmos-2 (Peng et al., 2023)) to extract a contextualized representation of the scene and task. The visual and textual inputs are tokenized and formed into a unified multimodal sequence. A learnable action token $\mathbf{t}_{act}$ is appended to it, and the corresponding output hidden state $\hat{\mathbf{t}}_{act} \in \mathbb{R}^{D_{act}}$, where $D_{act}$ represents the feature dimension, is extracted as the semantic action representation, encapsulating task-oriented behavior grounded in multi-modal context.

In parallel, the ESM processes a third-view image $I_t^{3rd}$, along with optional geometric inputs such as depth $D_t$ and camera poses $P$, to extract spatial structure representations. Through a spatial encoder $\mathcal{E}_{spl}(\cdot)$, it outputs a set of spatial tokens $\mathbf{T}_{spl}$, encoding global 3D geometric priors essential for scene understanding. Further details are provided in Sec. 2.4.

The extracted semantic action token $\hat{\mathbf{t}}_{act}$ and spatial tokens $\mathbf{T}_{spl}$ are then integrated in the Spatial-Enhanced Action Head, collectively guide action generation. We introduce a lightweight fusion

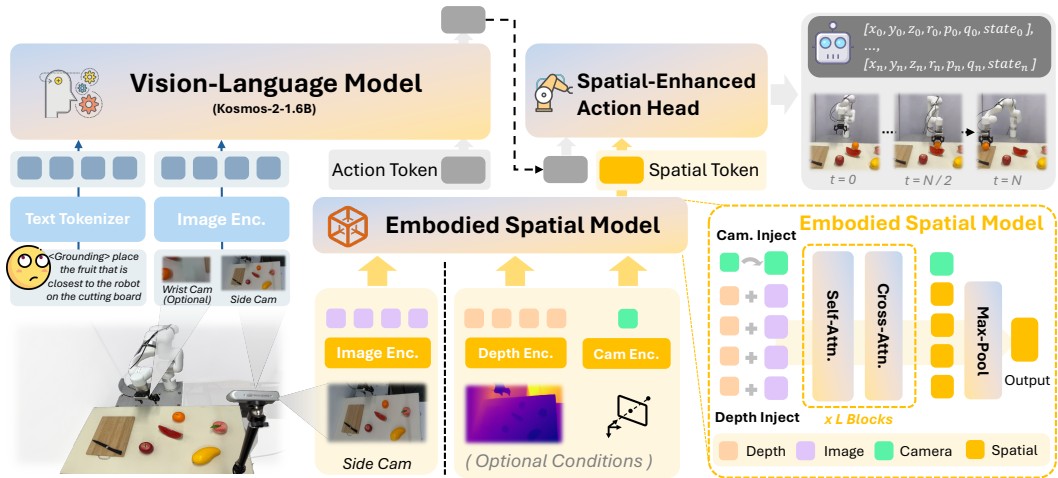

Figure 2: **Overview of FALCON framework.** FALCON integrates a 2D VLM (e.g., Kosmos-2), an Embodied Spatial Model, and a Spatial-Enhanced Action Head. At timestep $t$, the VLM processes visual observations $O_t$ and language instructions $L$ to produce a semantic action token $\hat{\mathbf{t}}_{\text{act}}$. Concurrently, the Embodied Spatial Model encodes a third-view image $I_t^{\text{3rd}}$ and optional geometric inputs into spatial tokens $\mathbf{T}_{\text{spl}}$. These are fused by the Spatial-Enhanced Action Head to generate precise robot actions $A_t$, enabling robust manipulation through joint semantic and spatial reasoning.

mechanism that aligns and combines these complementary representations (see Sec. 2.5 for more details), serving as input to an action predictor that outputs action sequences $A_t$. This novel design ensures that both high-level semantic context and 3D structural representation are directly incorporated into the policy, significantly improving precision and generalization in manipulation tasks.

## 2.3 TRAINING OBJECTIVE

During the training process of FALCON, the objective for action sequence generation is formulated as the minimization of a composite loss function over the predicted action horizon. Specifically, we compute the discrepancy between the predicted action sequence $a_{t:t+C-1}$ and the corresponding ground truth sequence $\hat{a}_{t:t+C-1}$ using two complementary loss components: the Mean Squared Error (MSE) and the Binary Cross-Entropy (BCE). The overall loss function is defined as:

$$\mathcal{L} = \sum_{i=t}^{t+C-1} \text{MSE}(\hat{a}_{i,\text{pose}}, a_{i,\text{pose}}) + \lambda \cdot \text{BCE}(\hat{a}_{i,\text{gripper}}, a_{i,\text{gripper}}), \qquad (2)$$

where the MSE term penalizes errors in the first six dimensions of the action vector and the BCE loss is applied to the last gripper dimension. The weighting factor $\lambda$ balances the contributions of the two loss terms, ensuring stable and representative learning across heterogeneous action components.

To integrate spatial awareness into the VLA model while preserving the pre-trained components' capabilities, we carefully design a two-stage post-training pipeline. Drawing inspiration from (Liu et al., 2023), in **Stage 1** we freeze all pre-trained components and optimize only a lightweight adapter to achieve initial alignment between spatial tokens and the VLA's feature space. Building upon this aligned foundation, **Stage 2** unfreezes both the VLM and adapter for joint refinement while keeping the remaining parts frozen, thereby enabling the VLM implicitly refines it's semantic features to incorporate spatial cues, which subsequently inform action prediction. The detailed training paradigms of the FALCON model are available in Appendix A.3.

## 2.4 EMBODIED SPATIAL MODEL

Although recently proposed Spatial Foundation Models (Wang et al., 2024; 2025a) have shown promising results in image-only reconstruction, they cannot exploit additional 3D modalities commonly available in robotics, such as RGB-D cameras and calibrated poses. To address this limitation, we propose an **Embodied Spatial Model** that injects 3D conditions (*i.e.*, depth, pose) to build more accurate spatial representations, enabling our action head to predict precise trajectories in space.

Specifically, given an input image $I_t$ at time $t$, we follow VGGT (Wang et al., 2025a) to encode it into spatial tokens $\mathbf{T}_{\text{spl}} \in \mathbb{R}^{M \times D_s}$, where $M$ is the token number per image and $D_s$ is the token dimension. The image is first tokenized into visual tokens $\mathbf{T}_{\text{vis}}$ via DINO (Oquab et al., 2024). These are then concatenated with a learnable camera token $\mathbf{t}_{\text{cam}} \in \mathbb{R}^{D_s}$ and fed into a Spatial Encoder $\mathcal{E}_{\text{spl}}(\cdot)$, which consists of $N$ cross-attention and self-attention blocks:

$$(\mathbf{T}_{\text{spl}}, \hat{\mathbf{t}}_{\text{cam}}) = \mathcal{E}_{\text{spl}}(\mathbf{T}_{\text{vis}}, \mathbf{t}_{\text{cam}}). \tag{3}$$

The resulting spatial tokens $\mathbf{T}_{\text{spl}}$ and refined camera tokens $\hat{\mathbf{t}}_{\text{cam}}$ are passed to a depth predictor and a camera predictor, respectively, to achieve accurate scene reconstruction. Notably, the spatial tokens $\mathbf{T}_{\text{spl}}$, which encapsulate rich spatial priors, have shown significant benefits for spatial understanding when integrated into VLMs (Fan et al., 2025; Wu et al., 2025).

**3D Conditions Encoding.** First, as shown in Fig. 2, given camera pose $P \in \mathbb{R}^7$ input, we encode the intrinsic and normalized extrinsic of the side camera into the GT camera token $\mathbf{t}_{\text{gt-cam}} \in \mathbb{R}^{D_s}$ using an MLP-based camera encoder $\mathcal{E}_{cam}(\cdot)$. Given depth $D_t \in \mathbb{R}^{H \times W}$ and its valid map $M_{\text{dpt}} \in \mathbb{R}^{H \times W}$ input, we first normalize depth $D'_t = D_t / \text{Norm}(D_t)$ to handle any depth ranges at train and test time. The normalized depth map $D'_t$ is concatenated with its corresponding validity mask $M_{\text{dpt}}$, enabling us to capture frame-wise incomplete depth information. The resulting tensor is then passed through a depth encoder $\mathcal{E}_{\text{dpt}}(\cdot)$, which consists of a stack of convolutional layers with a kernel size $14 \times 14$, effectively partitioning it into a sequence of tokens $\mathbf{T}_{\text{dpt}}$ that are aligned in size with the image tokens $\mathbf{T}_{\text{vis}}$. The formulation is shown below:

$$\mathbf{t}_{\text{gt-cam}} = \mathcal{E}_{cam}(\mathbf{t}_{\text{cam}}), \quad \mathbf{T}_{\text{dpt}} = \mathcal{E}_{\text{dpt}}([D'_t \,\|\, M_{\text{dpt}}]), \quad \mathbf{T}_{\text{dpt}} \in \mathbb{R}^{M \times D_s}, \tag{4}$$

where $[\cdot \| \cdot]$ denotes channel-wise concatenation.

**3D Conditions Injection and Training Strategy.** After obtaining the GT pose token $\mathbf{t}_{\text{gt-cam}}$ and depth tokens $\mathbf{T}_{\text{dpt}}$, our objective is not only to achieve accurate reconstruction through the reconstruction head under 3D-conditioned settings, but also to enhance the geometric grounding ability of the spatial tokens $\mathbf{T}_{\text{spl}}$ generated by the Spatial Encoder. To this end, we replace the learnable camera token $\mathbf{t}_{\text{cam}}$ with the GT camera token $\mathbf{t}_{\text{gt-cam}}$, and fuse the depth tokens $\mathbf{T}_{\text{dpt}}$ with the image tokens $\mathbf{T}_{\text{vis}}$ via element-wise addition. Meanwhile, in robotic applications, depth sensors or accurate camera poses may not always be available (e.g., Open X-Embodiment dataset). To preserve the model's ability to reason about spatial structure even without 3D conditions, we design a stochastic conditioning strategy. Specifically, we randomly decide whether to inject depth and/or pose during training, formulated as:

$$(\mathbf{T}_{\text{spl}}, \hat{\mathbf{t}}_{\text{cam}}) = \mathcal{E}_{\text{spl}}\Big(\mathbf{T}_{\text{vis}} + b_{\text{d}}\mathbf{T}_{\text{dpt}}, \; b_{\text{p}}\mathbf{t}_{\text{gt-cam}} + (1 - b_{\text{p}})\mathbf{t}_{\text{cam}}\Big), \tag{5}$$

where $b_{\text{d}}, b_{\text{p}} \sim \text{Bernoulli}(p)$. This strategy ensures the model can exploit depth and pose cues when available, while retaining strong image-only spatial reasoning when they are absent. As for supervision, we follow VGGT (Wang et al., 2025a) to adopt depth, point map, and pose losses to formulate multi-task supervision.

## 2.5 SPATIAL-ENHANCED ACTION HEAD

As illustrated in Fig. 2, the proposed **Spatial-Enhanced Action Head** integrates geometric representations $\mathbf{T}_{\text{spl}}$ from the ESM with semantic features $\hat{\mathbf{t}}_{\text{act}}$ from the VLM, enabling more accurate and spatially-aware policy learning.

**Modality Fusion Strategies.** To combine these complementary representations, we first compress the spatial tokens $\mathbf{T}_{\text{spl}}$ into a unified vector $\mathbf{t}_{\text{spl}} \in \mathbb{R}^{D_s}$ through a max-pooling operation, then project it into the VLM's feature space to obtain $\widetilde{\mathbf{t}}_{\text{spl}}$ using a lightweight MLP adapter $\mathcal{D}$. The aligned spatial feature $\widetilde{\mathbf{t}}_{\text{spl}}$ is then fused with the semantic action token $\hat{\mathbf{t}}_{\text{act}}$ through highly efficient element-wise addition (see Sec. 3.3 for detailed ablations about different fusion strategies). The overall fusion process is formulated as:

$$\widetilde{\mathbf{t}}_{\text{spl}} \in \mathbb{R}^{D_{\text{act}}} = \mathcal{D}(\mathbf{t}_{\text{spl}}), \quad \mathbf{f}_{\text{fused}} = \hat{\mathbf{t}}_{\text{act}} + \widetilde{\mathbf{t}}_{\text{spl}}. \tag{6}$$

This dedicated fusion mechanism preserves the pre-trained representation space and generalizable capabilities of the VLM while enriching VLA with geometrically grounded structural awareness.

**Action Predictor.** The fused feature vector is forwarded to an action predictor $\pi$ to generate robot actions. We explore two distinct architectures for this predictor: An MLP-based predictor directly

maps the current fused feature vector to an action output: $A_t = \pi(\mathbf{f}_{\text{fused}}^t)$. For long-horizon robotic tasks that involve sequential decision-making, we employ a predictor based on the long short-term memory (LSTM) network (Hochreiter & Schmidhuber, 1997; Chung et al., 2014) that utilizes a history of feature representations. This approach processes the sequence $\mathbf{f}_{\text{fused}}^{t-H+1}, \ldots, \mathbf{f}_{\text{fused}}^t$ through the LSTM network, followed by an MLP that produces the final action chunk prediction: $A_t = \pi(\mathbf{f}_{\text{fused}}^{t-H+1}, \ldots, \mathbf{f}_{\text{fused}}^t)$, where $H$ denotes the history length. Each $\mathbf{f}_{\text{fused}}^i$ is obtained through the same feature fusion process described previously.

By integrating spatially rich tokens from the ESM with semantically grounded features from VLM, our model achieves enhanced spatial perception capabilities while retaining strong semantic alignment with language instructions. Moreover, the proposed fusion strategy significantly enhances the performance of the real-world spatial awareness tasks of the policy, as demonstrated in Sec. 3.2.

## 3 EXPERIMENTS

**Benchmarks.** For simulation, we evaluate on the widely used benchmarks *CALVIN* (Mees et al., 2022) and *SimplerEnv* (Li et al., 2024c). For real-world tasks, we design settings that span from simple interactions (e.g., lifting a yellow pepper) to long-horizon, spatially demanding activities (e.g., placing a red coke can on the bottom shelf), thereby thoroughly testing robustness and spatial reasoning. Further benchmark details are provided in Appendix A.6.

**Implementation Details.** FALCON is built on a Kosmos-2 (Peng et al., 2023) VLM backbone (~1.6B parameters), combined with a 1.0B-parameter Embodied Spatial Model and a Spatial-Enhanced Action Head, totaling 2.9B parameters. Training was conducted on a cluster of 32 A100 GPUs. Additional training and deployment details are available in Appendix A.4 and A.5.

### 3.1 SIMULATION EXPERIMENTS

**CALVIN Evaluations.** Tab. 1 presents the evaluation results on the CALVIN benchmark. Our method achieves SOTA performance in both the ABC→D and ABCD→D settings, significantly outperforming all prior approaches. These results highlight FALCON's strong ability to tackle diverse tasks and execute long-horizon, language-conditioned manipulation. Notably, in the challenging zero-shot ABC→D setting, FALCON surpasses previous methods that rely on ground-truth point clouds (e.g., 3DDP (Ze et al., 2024) and 3D Diffuser Actor (Ke et al., 2024)), improving the *Avg. Len.* by 4.13 and 1.05, respectively. This provides clear evidence of the effectiveness of our implicit spatial information integration strategy.

Table 1: **Long-horizon robotic manipulation evaluation on the CALVIN benchmark.**

| Method | Task | Tasks Completed in a Row (%) | | | | | Avg. Len. ↑ |
|---|---|---|---|---|---|---|---|
| | | 1 | 2 | 3 | 4 | 5 | |
| MCIL (Lynch & Sermanet, 2020) | ABCD→D | 37.3 | 2.7 | 0.2 | 0.0 | 0.0 | 0.40 |
| RT-1 (Brohan et al., 2022) | ABCD→D | 84.4 | 61.7 | 43.8 | 32.3 | 22.7 | 2.45 |
| Robo-Flamingo (Li et al., 2024b) | ABCD→D | 96.4 | 89.6 | 82.4 | 74.0 | 66.0 | 4.09 |
| GR-1 (Wu et al., 2024) | ABCD→D | 94.9 | 89.6 | 84.4 | 78.9 | 73.1 | 4.21 |
| UP-VLA (Zhang et al., 2025) | ABCD→D | 96.2 | 92.1 | 87.9 | 84.2 | 81.2 | 4.42 |
| RoboVLM (Li et al., 2024a) | ABCD→D | 96.7 | 93.0 | 89.9 | 86.5 | 82.6 | 4.49 |
| **FALCON (ours)** | ABCD→D | **97.2** | **93.3** | **90.3** | **88.0** | **84.0** | **4.53** |
| 3DDP (Ze et al., 2024) | ABC→D | 28.3 | 2.3 | 0.0 | 0.0 | 0.0 | 0.27 |
| MCIL (Lynch & Sermanet, 2020) | ABC→D | 30.4 | 1.3 | 0.2 | 0.0 | 0.0 | 0.31 |
| RT-1 (Brohan et al., 2022) | ABC→D | 53.3 | 22.2 | 9.4 | 3.8 | 1.3 | 0.90 |
| Robo-Flamingo (Li et al., 2024b) | ABC→D | 82.4 | 61.9 | 46.6 | 33.1 | 23.5 | 2.47 |
| GR-1 (Wu et al., 2024) | ABC→D | 85.4 | 71.2 | 59.6 | 49.7 | 40.1 | 3.06 |
| 3D Diffuser Actor (Ke et al., 2024) | ABC→D | 93.8 | 80.3 | 66.2 | 53.3 | 41.2 | 3.35 |
| UP-VLA (Zhang et al., 2025) | ABC→D | 92.8 | 86.5 | 81.5 | 76.9 | 69.9 | 4.08 |
| RoboVLM (Li et al., 2024a) | ABC→D | 98.0 | 93.6 | 85.4 | 77.8 | 70.4 | 4.25 |
| Seer-Large (Tian et al., 2024) | ABC→D | 96.3 | 91.6 | 86.1 | 80.3 | 74.0 | 4.28 |
| **FALCON (ours)** | ABC→D | **98.4** | **94.5** | **88.6** | **82.5** | **75.5** | **4.40** |

Table 2: **SimplerEnv evaluation across different policies on WidowX Robot tasks. Put Spoon**: Put Spoon on Towel. **Put Carrot**: Put Carrot on Plate. **Stack Block**: Stack Green Block on Yellow Block. **Put Eggplant**: Put Eggplant in Yellow Basket.

| Method | Put Spoon | Put Carrot | Stack Block | Put Eggplant | Average |
|---|---|---|---|---|---|
| RT-1-X (O'Neill et al., 2024) | 0.0% | 4.2% | 0.0% | 0.0% | 1.1% |
| OpenVLA (Kim et al., 2024) | 0.0% | 0.0% | 0.0% | 4.1% | 1.0% |
| Octo-Base (Octo Model Team et al., 2024) | 12.5% | 8.3% | 0.0% | 43.1% | 16.0% |
| RoboVLM (Li et al., 2024a) | 45.8% | 20.8% | 4.2% | 79.2% | 37.5% |
| SpatialVLA (Qu et al., 2025) | 16.7% | 25.0% | **29.2%** | 100.0% | 42.7% |
| **FALCON (ours)** | **62.5%** | **41.7%** | 20.8% | **100.0%** | **56.3%** |

Table 3: **SimplerEnv evaluation across different policies on Google Robot tasks. Open/Close**: Open / Close Drawer. **Drawer Apple**: Open Top Drawer and Place Apple.

| Method | Pick Coke Can | Move Near | Open/Close | Drawer Apple | Average |
|---|---|---|---|---|---|
| RT-1-X (O'Neill et al., 2024) | 56.7% | 31.7% | **59.7%** | 21.3% | 42.4% |
| RT-2-X (O'Neill et al., 2024) | 78.7% | 77.9% | 25.0% | 3.7% | 46.3% |
| Octo-Base (Octo Model Team et al., 2024) | 17.0% | 4.2% | 22.7% | 0.0% | 11.0% |
| OpenVLA (Kim et al., 2024) | 16.3% | 46.2% | 35.6% | 0.0% | 24.5% |
| TraceVLA (Zheng et al., 2024) | 28.0% | 53.7% | 57.0% | 0.0% | 34.7% |
| RoboVLM (Li et al., 2024a) | 77.3% | 61.7% | 43.5% | 24.1% | 51.7% |
| SpatialVLA (Qu et al., 2025) | 86.0% | 77.9% | 57.4% | 0.0% | 55.3% |
| **FALCON (ours)** | **90.7%** | **79.2%** | 39.8% | **41.7%** | **62.9%** |

**SimplerEnv Evaluations.** Tab. 2 reports the results on the Bridge-WidowX setup, where FALCON consistently outperforms all baselines and achieves best performance. The notable improvements are observed in challenging tasks like *Put Spoon on Towel* (16.7% vs. 62.5%) and *Put Carrot on Plate* (25.0% vs. 41.7%), demonstrating FALCON's superior adaptability and effectiveness.

Tab. 3 summarizes the performance of various generalist policies on the Google Robot setup. FALCON achieves an overall success rate of 62.9%, surpassing all baseline methods. Notably, on the challenging task *Open Top Drawer and Place Apple*, most baselines show near-zero success rates. Even the large-scale closed-source model RT-2-X (O'Neill et al., 2024) with 55B parameters achieves only 3.7% success, while FALCON delivers an impressive 41.7%, highlighting its exceptional generalization and spatial perception capabilities.

## 3.2 REAL-WORLD EXPERIMENTS

To enable a more comprehensive evaluation, we conduct a series of carefully designed real-world experiments covering diverse object manipulation scenarios with varying task variations. The experiments are organized into three distinct settings: *Base Tasks*, *Few-shot Adaptation*, and *Spatial Understanding Capability Evaluations*. All models are initially pre-trained on a mixture of the Open X-Embodiment dataset (O'Neill et al., 2024) and then fine-tuned with multi-task real-robot data. Relevant qualitative results are provided in Appendix A.11.

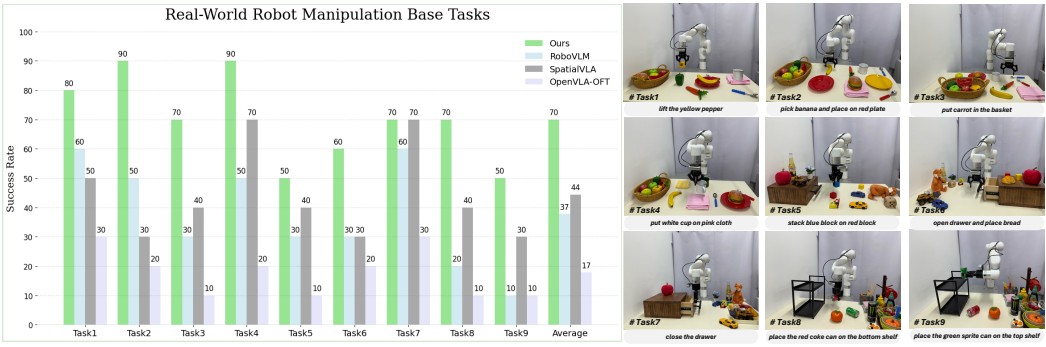

Figure 3: **Evaluation of base tasks in cluttered scene.** *Base Tasks* contains a total of nine distinct task suites, encompassing language grounding (cluttered scenes with random distractors) and semantic understanding (unseen object poses).

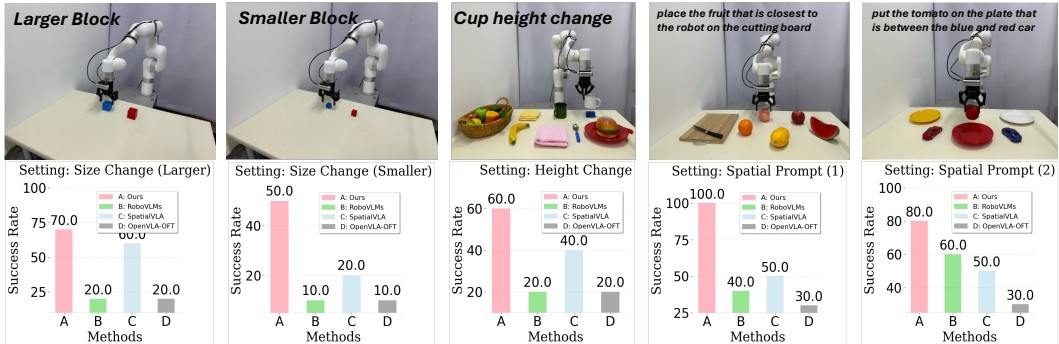

Figure 5: **Spatial Understanding Capability Evaluations** consist of four tasks with varying levels of spatial complexity, designed to further investigate the spatial perception capabilities of FALCON.

**Base Tasks.** As shown in Fig. 3, FALCON achieves the highest average success rate of 70.0% across all nine task suites, outperforming the advanced method SpatialVLA (Qu et al., 2025) (44.4%) by 25.6%. Moreover, in the task *pick banana and place on red plate*, while RoboVLM (Li et al., 2024a) and OpenVLA-OFT (Kim et al., 2025) often erroneously place the banana onto a yellow plate, FALCON consistently places it on the correct red plate, demonstrating precise instruction following capability and superior scene understanding.

**Few-shot Adaptation.** As shown in Fig. 4, FALCON achieves the highest performance across all settings, significantly outperforming the second-best model by 27.5% in *Simple* and 27% in *Unseen Average*. Notably, in the *Unseen Object* variation of the task *open drawer and place bread* (Fig. 1), FALCON achieves an impressive success rate of 80%, while other models demonstrate near-zero success. Success rates for individual variants and sub-tasks are provided in Appendix A.11.2.

**Spatial Understanding Capability Evaluations.** As illustrated in Fig. 5, FALCON demonstrates superior spatial understanding, outperforming all existing policies across the evaluated tasks. Notably, baseline methods such as RoboVLM often struggle with objects of varying sizes. For larger blocks, collisions frequently occur during the placement of the blue block, while smaller blocks are prematurely released before placement, leading to task failure. In contrast, our method exhibits strong robustness to scale variations, avoiding these issues and achieving the highest success rates in both scenarios.

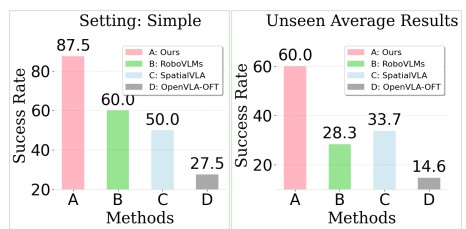

Figure 4: **Performance comparison of different methods**. *Simple* setting (left): includeing four challenging tasks selected from *Base Tasks*. *Unseen* scenarios (right): containing three unseen variations: *Unseen Object, Background*, and *Task Description* to evaluate the robustness and generalization of all models.

## 3.3 IN-DEPTH ANALYSIS

Table 4: **Ablation studies on spatial token injection methods and fusion strategies.** Results confirm that the standard FALCON paradigm achieves the best performance, validating it as the optimal design choice.

| Method | Task | Tasks Completed in a Row (%) | | | | | Avg. Len. ↑ |
|---|---|---|---|---|---|---|---|
| | | 1 | 2 | 3 | 4 | 5 | |
| FALCON_VLM-tokens | ABCD→D | 92.9 | 85.4 | 79.4 | 74.4 | 68.1 | 4.00 |
| Cross-Attention | ABCD→D | 93.7 | 85.5 | 78.2 | 73.0 | 67.5 | 3.98 |
| FiLM-Gated | ABCD→D | 93.8 | 85.7 | 80.2 | 75.4 | 69.6 | 4.04 |
| **FALCON (ours)** | ABCD→D | **94.0** | **86.7** | **80.8** | **76.4** | **70.9** | **4.08** |
| FALCON_VLM-tokens | ABC→D | 94.2 | 85.2 | 75.6 | 66.1 | 57.6 | 3.79 |
| Cross-Attention | ABC→D | 91.3 | 81.9 | 72.9 | 64.9 | 57.2 | 3.68 |
| FiLM-Gated | ABC→D | 92.9 | 83.7 | 74.5 | 66.9 | 58.4 | 3.76 |
| **FALCON (ours)** | ABC→D | **93.7** | **86.9** | **77.9** | **70.3** | **62.2** | **3.91** |

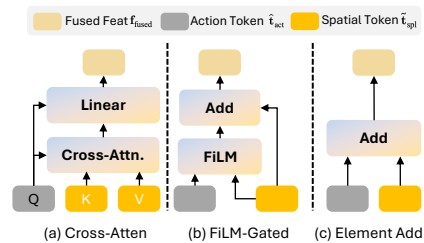

Figure 6: **Different modality fusion strategies between spatial and semantic action tokens.**

**Spatial Token Injection.** To verify the effectiveness of our strategy for injecting 3D information into the action head, we evaluate a variant following the approach of most 3D-based VLAs, where spatial tokens from the ESM are directly injected into the VLMs (denoted as FALCON$_{\text{VLM-tokens}}$) on CALVIN. As shown in Tab. 4, this approach results in significant performance degradation compared to the standard FALCON paradigm. Notably, in the zero-shot ABC→D setting, the *Avg. Len.* drops significantly from 3.91 to 3.79. These results indicate that introducing fine-grained spatial features into the VLM disrupts its pre-trained semantic representation space, negatively impacting its generalization capability. In contrast, injecting spatial tokens directly into the action head preserves the VLM's integrity while effectively utilizing geometric cue, making it the superior strategy.

**Modality Fusion Strategies.** To combine the aligned spatial feature $\widetilde{t}_{\text{spl}}$ with the semantic action token $\hat{t}_{\text{act}}$, we further investigate three distinct fusion approaches:

1) *Cross-Attention Fusion:* The action token $\hat{t}_{\text{act}}$ serves as the query, while the projected spatial feature $\widetilde{t}_{\text{spl}}$ provide key and value inputs. Multi-head attention enables adaptive feature recalibration based on cross-modal relevance.

2) *FiLM-Gated Modulation:* This method uses the spatial feature $\widetilde{t}_{\text{spl}}$ to generate affine parameters $(\gamma, \beta)$ for feature-wise linear modulation of the action token $\hat{t}_{\text{act}}$, followed by a gating mechanism that learns to blend the modulated semantic and original spatial features.

3) *Element-wise Addition:* A direct, non-parametric combination of the two feature vectors.

As shown in Tab. 4, element-wise addition (our standard FALCON) consistently delivers the best performance across all experimental settings. This method achieves the highest task success rates while remaining both simple and computationally efficient, as it introduces no additional parameters. The results underscore the effectiveness of a straightforward, parameter-free fusion strategy for combining spatial and semantic representations in VLA models.

Table 5: **Performance comparison of different modality input on CALVIN benchmark.** Kosmos-VLA (*w/ rgb*) is a 2D VLA without ESM. Kosmos-VLA (*w/ rgb-d*) is a point cloud-based variant where the ESM is replaced by a lightweight point cloud encoder (Ze et al., 2025) while retaining other parts.

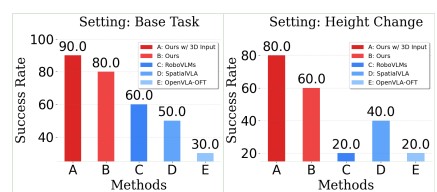

Figure 7: **Performance comparison of different modality input on real-world tasks. Left**: *lift the yellow pepper*. **Right**: *put white cup on pink cloth– cup height change.*

| Method | Task | Tasks Completed in a Row (%) | | | | | Avg. Len. ↑ |
|---|---|---|---|---|---|---|---|
| | | 1 | 2 | 3 | 4 | 5 | |
| Kosmos-VLA (*w/ rgb*) | ABCD→D | 92.5 | 85.4 | 79.1 | 74.9 | 68.7 | 4.01 |
| Kosmos-VLA (*w/ rgb-d*) | ABCD→D | 92.4 | 85.3 | 80.0 | 76.5 | 70.5 | 4.05 |
| FALCON (*w/ rgb*) | ABCD→D | **94.0** | 86.7 | 80.8 | 76.4 | 70.9 | 4.08 |
| FALCON (*test w/ d*) | ABCD→D | 93.8 | 86.6 | **81.8** | 76.6 | **71.3** | **4.09** |
| FALCON (*w/ rgb-d*) | ABCD→D | **94.0** | **87.0** | 81.3 | **76.8** | 70.3 | **4.09** |
| FALCON (*test w/o d*) | ABCD→D | 93.7 | 86.4 | 80.9 | **76.8** | 69.8 | 4.07 |
| Kosmos-VLA (*w/ rgb*) | ABC→D | 91.6 | 80.0 | 68.3 | 58.8 | 49.0 | 3.48 |
| Kosmos-VLA (*w/ rgb-d*) | ABC→D | 93.6 | 86.0 | 78.6 | **73.3** | **66.3** | 3.98 |
| FALCON (*w/ rgb*) | ABC→D | 93.7 | 86.9 | 77.9 | 70.3 | 62.2 | 3.91 |
| FALCON (*test w/ d*) | ABC→D | 94.2 | **88.4** | **79.4** | 70.9 | 61.8 | 3.95 |
| FALCON (*w/ rgb-d*) | ABC→D | **94.7** | 86.7 | 79.1 | 72.4 | 64.4 | 3.97 |
| FALCON (*test w/o d*) | ABC→D | 94.0 | 87.0 | 78.6 | 71.4 | 63.8 | 3.95 |

**Modality Transferability.** To evaluate the modality transferability of FALCON, we conduct extensive experiments on both the CALVIN benchmark and real-world tasks to demonstrate the benefits of additional modality inputs for our approach. As shown in Tab. 5, under identical input conditions, FALCON outperforms Kosmos-VLA in the ABCD→D setting. In the zero-shot ABC→D setting, both methods achieve comparable performance under RGB-D input. Furthermore, FALCON using only RGB input achieves competitive performance compared to Kosmos-VLA with RGB-D input and surpasses it with same input modality by 0.43 in *Avg. Len.*.

Besides, we conduct experiments under varying test-time depth conditions while keeping the training modality fixed. Specifically, we examine two scenarios: (1) model trained only on RGB but evaluate with additional depth input (*test w/ d*), and (2) model trained on RGB-D but evaluate without depth (*test w/o d*). As summarized in Tab. 5, under ABCD→D setting, providing depth at test-time improves the *Avg. Len.* from 4.08 to 4.09. Conversely, when depth is omitted at test-time for model trained with RGB-D, performance remains robust. A similar trend is observed in the more challenging ABC→D setting: RGB-only FALCON benefits from test-time depth input, as *Avg. Len.* increases from 3.91 to 3.95, matching the performance of model trained with full RGB-D.

Table 6: **Zero-shot monocular depth estimation with diverse input modalities for ESM on CALVIN benchmark.** $\delta < 1.25$: measures the accuracy under 1.25 threshold (percentage of predictions with relative error $\leq 25\%$). Abs. Rel: computes the average absolute relative error between prediction and ground truth.

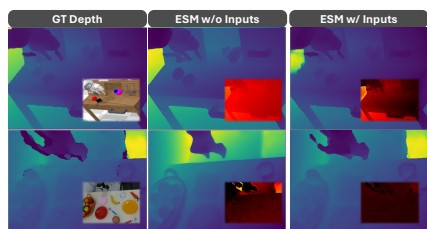

| Method | Depth | Camera | $\delta < 1.25$ (%) ↑ | Abs. Rel ↓ |
|---|---|---|---|---|
| VGGT (Wang et al., 2025a) | - | - | 91.33 | 8.53 |
| **Ours** | ✗ | ✗ | 90.91 | 8.61 |
| **Ours** | ✓ | ✗ | **99.79** | 0.91 |
| **Ours** | ✓ | ✓ | 99.47 | **0.87** |

Figure 8: **Depth map Visualization.**

Real-world experiments further validate that incorporating depth and camera poses significantly enhances FALCON's robustness (Fig. 7), increasing task success rates from 60% to 80% in scenarios involving objects of varying heights. These findings highlight FALCON's effective utilization of additional geometric information and its adaptability across different sensory modalities.

**Embodied Spatial Model**. To further investigate the role of ESM, we conduct additional zero-shot experiments on CALVIN to evaluate the monocular depth estimation results of ESM. These experiments further validate why ESM effectively leverages additional modality inputs to achieve performance gains. As shown in Tab. 6, our ESM achieves performance comparable to VGGT when using only RGB input. Moreover, its performance improves significantly when additional depth or camera pose information is accessible. This demonstrates the inherent strength of FALCON's modality transferability, which stems from the ability of our proposed ESM to seamlessly benefit from diverse 3D modality inputs. Fig. 8 presents ESM predicted depth maps and corresponding error maps (darker colors indicate smaller errors) under different input modalities.

## 4 RELATED WORKS

**3D-Enhanced Vision-Language-Action Models**. While VLAs (Brohan et al., 2023; Kim et al., 2024; Li et al., 2024b;a; Cheang et al., 2024; Black et al., 2024; Kim et al., 2025; Cheang et al., 2025; Zhang et al., 2025; Su et al., 2025a;b) have advanced generalist robotics by leveraging pretrained VLMs, their 2D nature limits spatial reasoning. Recent works address this by integrating 3D cues through two main approaches: (1) explicit 3D inputs like point clouds (Li et al., 2025; Sun et al., 2025), which require specific sensors and lack modality transferability; or (2) embedding 3D features into VLM input space (Zhen et al., 2024; Qu et al., 2025), which often disrupts pre-trained alignments and requires costly retraining. FALCON overcomes these limitations by decoupling spatial processing from the VLM, preserving semantic integrity while ensuring strong modality transferability.

**Spatial Foundation Models**. Recent advancements in deep learning have introduced novel alternatives to traditional SfM methods. Recently, VGGT (Wang et al., 2025a) proposes a multi-view architecture that processes multiple images simultaneously, moving beyond pairwise processing to improve reconstruction consistency and robustness. However, their integration into generalist robot policies remains challenging, often requiring complex feature alignment or suffering from information loss when adapted to action spaces. See comprehensive review of related works in Appendix A.2

## 5 CONCLUSIONS

In this work, we introduce FALCON, a vision-language-action model that augments generalist robot policies with robust 3D spatial understanding. FALCON makes three main contributions: (1) integration of spatial tokens from foundation models to provide strong geometric priors; (2) an Embodied Spatial Model that optionally incorporates 3D modalities (e.g., depth, camera poses) while preserving RGB-only functionality; and (3) a Spatial-Enhanced Action Head that injects spatial tokens directly into the control policy, avoiding disrupting vision-language alignment within the VLM. Experiments across both simulation and real-world tasks show that FALCON consistently surpasses existing VLA methods, achieving state-of-the-art performance and robustness on spatially demanding tasks.

## ACKNOWLEDGEMENTS

We would like to thank all members at ByteDance Seed Robotics team for their continuous support throughout this project. We also want to extend our gratitude to Wenke Xia for his essential assistance with the robotic hardware setup, to Haosong Peng for his valuable discussions, and to Hang Li for his leadership of this team.

## ETHICS STATEMENT

This work focuses on improving spatial reasoning for robotic manipulation. While our model is trained on standard open-source datasets and tested in controlled settings, we acknowledge that any AI system may potentially exhibit biases or produce unexpected behaviors. Our research is intended for academic exploration only, and we emphasize that any such outcomes do not reflect the views of the authors. We support the development of AI technologies that are ethical, safe, and aligned with societal values.

## REPRODUCIBILITY STATEMENT

We detail our work in the Methodology section (Sec. 2) and describe implementation details at the beginning of Sec. 3 and Appendix A.3, A.4, A.5, A.6.

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

# A  APPENDIX

## A.1  DECLARATION OF LLM USAGE

In the preparation of this work, the authors used LLM (e.g., GPT-4) in order to improve the readability and language of the manuscript. After using this tool, the authors reviewed and edited the content as needed and take full responsibility for the content of the published article.

## A.2  RELATED WORKS

### A.2.1  3D-ENHANCED VISION-LANGUAGE-ACTION MODELS

The development of generalist robot policies has been significantly advanced by VLAs (Brohan et al., 2023; Kim et al., 2024; Li et al., 2024b;a; Cheang et al., 2024; Black et al., 2024; Kim et al., 2025; Cheang et al., 2025; Zhang et al., 2025; Su et al., 2025a;b), which leverage large-scale pre-trained VLMs to connect visual and linguistic understanding with action generation. For instance, RT-2 (Brohan et al., 2023) and OpenVLA (Kim et al., 2024) fine-tune VLMs on robot data, representing actions as language tokens. While demonstrating impressive instruction-following capabilities, these methods operate primarily in the 2D domain, lacking explicit mechanisms for 3D geometric perception, which is a critical limitation in tasks requiring precise spatial understanding. To address this, recent works have integrated 3D perceptual cues into policy learning. One line of research directly consumes explicit 3D representations like point clouds for action prediction, as seen in PointVLA (Li et al., 2025) and GeoVLA (Sun et al., 2025). While enhancing geometric awareness, these methods rely on explicit 3D inputs and specific sensor configurations, limiting modality transferability when such inputs are unavailable. An alternative strategy embeds 3D features into the VLM's input space, exemplified by 3D-VLA (Zhen et al., 2024), SpatialVLA (Qu et al., 2025), and Evo-0 (Lin et al., 2025), but this often disrupts pre-trained vision-language alignment, necessitating costly embodied instruction tuning to recover performance. FALCON overcomes these issues by decoupling spatial processing from the VLM, preserving its semantic integrity while achieving strong modality transferability and robust performance across diverse sensory conditions, reducing dependency on specialized sensor setups when deploy in the real-world.

### A.2.2  SPATIAL FOUNDATION MODELS

Recent advancements in deep learning have introduced novel alternatives to traditional SfM methods. DUSt3R (Wang et al., 2024) represents a significant deviation from conventional SfM pipelines by predicting point clouds from image pairs without relying on geometric constraints or inductive biases. Unlike traditional SfM, which depends on keypoint matching and geometric optimization, DUSt3R generates predictions in a shared coordinate frame, enabling robust reconstruction across diverse scenes. This approach addresses several challenges inherent in classical methods, such as sensitivity to initialization and sparse correspondences. Building on this paradigm, several works have proposed variations with distinct architectural innovations. MASt3R (Leroy et al., 2024) improves the estimation of the pixel-wise correspondence between image pairs, strengthening the efficacy of unconstrained feed-forward models for SfM tasks. CUT3R (Wang et al., 2025b) introduces a recurrent formulation of DUSt3R, achieving computational efficiency at the expense of marginal accuracy degradation. More recently, VGGT (Wang et al., 2025a) proposes a multi-view architecture that processes multiple images simultaneously, moving beyond pairwise processing to improve reconstruction consistency and robustness. However, their integration into generalist robot policies remains challenging, often requiring complex feature alignment or suffering from information loss when adapted to action spaces.

## A.3  TRAINING PARADIGM

### A.3.1  FALCON TRAINING PARADIGM

Rather than incorporating spatial information during pre-training, which would significantly increase computational cost and complicate optimization, we adopt a post-training approach that preserves the original pre-training efficiency while enabling efficient integration of 3D geometric cues.

Let $\Theta_V$, $\Theta_A$, $\Theta_G$, and $\Theta_D$ denote the parameter sets of the VLM $\mathcal{V}$, action predictor $\pi$, ESM $\mathcal{G}$, and lightweight adapter $\mathcal{D}$, respectively. The overall training objective is to minimize the expected action prediction loss $\mathcal{L}$ (refer to Eq. 2) over the target data distribution:

**Stage 1: Feature Space Alignment.** In this initial stage, we freeze all pre-trained components ($\Theta_V$, $\Theta_A$, $\Theta_G$) and optimize only the adapter parameters $\Theta_D$. The adapter architecture employs a zero-initialized final linear layer, which ensures the spatial tokens $\mathbf{T}_{\text{spl}}$ initially contribute minimally to the fused representation, thereby preserving the pre-trained feature space and ensuring stable optimization. The training objective is:

$$\min_{\Theta_D} \mathbb{E}_{(O_t,L,\hat{A}_t)\sim\mathcal{S}} \left[ \mathcal{L}\left( \hat{A}_t, \pi\left( \mathcal{V}(O_t, L) + \mathcal{D}(\text{MaxPooling}(\mathcal{G}(I_t^{\text{3rd}}))) \right) \right) \right], \tag{7}$$

where $\mathbb{E}_{(O_t,L,\hat{A}_t)\sim\mathcal{S}}$ denotes the expectation over the dataset $\mathcal{S}$ of image-language-action tuples. This stage facilitates gradual alignment of spatial tokens with the semantic feature space without disrupting pre-trained representations.

**Stage 2: Joint Feature Refinement.** Building upon the aligned features from **Stage 1**, we unfreeze both the VLM parameters $\Theta_V$ and adapter parameters $\Theta_D$, while keeping $\Theta_A$ and $\Theta_G$ frozen. This allows the VLM to adapt its feature representation to effectively incorporate spatial information while maintaining linguistic understanding. The optimization objective becomes:

$$\min_{\Theta_V,\Theta_D} \mathbb{E}_{(O_t,L,\hat{A}_t)\sim\mathcal{S}} \left[ \mathcal{L}\left( \hat{A}_t, \pi\left( \mathcal{V}(O_t, L) + \mathcal{D}(\text{MaxPooling}(\mathcal{G}(I_t^{\text{3rd}}))) \right) \right) \right]. \tag{8}$$

This phased approach ensures stable convergence and prevents the spatial features from overwhelming the semantic representations during initial learning phases.

The proposed training strategy offers several advantages: (1) it maintains the integrity of the pre-trained VLA's knowledge while incorporating new spatial capabilities; (2) zero-initialization in **Stage 1** ensures training stability and avoids disruptive feature shifts; (3) gradual unfreezing in **Stage 2** enables balanced feature adaptation. This paradigm allows FALCON to achieve robust 3D spatial perception while maintaining strong performance across diverse tasks and scenarios.

### A.3.2 EMBODIED SPATIAL MODEL TRAINING PARADIGM

ESM is trained following the dataset and preprocessing procedures of VGGT (Wang et al., 2025a). For each training batch, we randomly sample 1–12 frames from a randomly selected scene, resulting in a total of 24 images per batch. Training is performed using the AdamW optimizer with differentiated learning rates: 1e-6 for the large unified transformer backbone and 1e-5 for the depth, camera, and point heads. We set Bernoulli probability $p = 66\%$ throughout ESM training. The complete training process requires 16 A100 GPUs and runs for approximately 2 days.

### A.4 HYPER-PARAMETERS AND TRAINING DETAILS

**Simulation Benchmarks.** As FALCON employs a two-stage post-training strategy, we first pre-train a 2D VLA (denotes as *Kosmos-VLA-2D*) on the target datasets (CALVIN (Mees et al., 2022), OXE dataset (O'Neill et al., 2024)) without involving the ESM. The pre-training uses a learning rate of 2e-5, a global batch size of 128, and a warmup ratio of 0.25 epochs for CALVIN and 2.5k steps for OXE. Subsequently, we conduct post-training based on the pre-trained weights of *Kosmos-VLA-2D* in two stages: **Stage 1** uses a learning rate of 1e-4, a global batch size of 128, and no warmup. **Stage 2** hyper-parameters are detailed in Tab. 7.

**Real-World Tasks.** For real-world evaluation, we initialize the model with pre-trained weights from *Kosmos-VLA-2D* (OXE) and ESM. **Stage 1** training uses a learning rate of 1e-4, a global batch size of 512, and no warmup. **Stage 2** hyper-parameters are provided in Tab. 7. We evaluate all VLAs under the following training settings: (1) *Base Tasks*: models trained separately on three scenarios. (2) *Few-shot Adaptation*: multi-task training across all four tasks. (3) *Spatial-Prompts*: efficient fine-tuning on two tasks. Baseline models (SpatialVLA (Qu et al., 2025), RoboVLM (Li et al., 2024a)) use the same hyper-parameters as FALCON. For OpenVLA-OFT (Kim et al., 2025) (LoRA-only), we use a learning rate of 5e-4, LoRA rank 32, chunk size 5, and *Base Tasks* train for 150k iterations, other two settings for 100k iterations. All VLAs trained on real-world datasets use only the side camera input.

Across all experiments, both training stages for FALCON use the same number of epochs/iterations, a *Constant* learning rate scheduler, and the *AdamW* optimizer. All input images (including depth maps) are resized to a resolution of 224×224. Besides, we follow (Li et al., 2024a) to set the loss weighting factor $\lambda = 0.01$ for fair comparisons.

**Checkpoint Selection.** Following the findings of (Li et al., 2024a), we note that policy performance does not fully correlate with offline evaluation metrics (e.g., validation loss) due to compounding errors in long-horizon rollouts, making checkpoint selection challenging. To ensure fair comparisons, all VLAs are trained for a fixed number of epochs or iterations. Specifically:

- On **CALVIN**, models are trained for 5 epochs with a batch size of 128 truncated trajectories, and the final checkpoint is evaluated.
- On **SimplerEnv**, models are trained for 150K iterations (batch size 128), with evaluation at 10K-interval checkpoints, and the best-performing model is reported.
- In **Real-World** experiments, all models are trained for 30 epochs or same iterations for OpenVLA-OFT with a batch size of 512 truncated trajectories, and only the final checkpoint is evaluated.

This consistent protocol ensures comparable evaluation across all baselines.

## A.5 IMPLEMENTATION DETAILS

As demonstrated in Tab. 7, on the CALVIN benchmark, the VLM receives side and wrist camera images with a history length of 16 frames, while the ESM processes third-view images also with same length historical context. Predictions are made using an LSTM-based action predictor that outputs a chunk of $C = 10$ future actions. For SimplerEnv and the real-world benchmark, both the VLM and ESM receive a single-step third-view image, and an MLP-based action predictor predicts an action chunk of length $C = 5$. During inference, we employ two different action execution strategies: for CALVIN and SimplerEnv, the policy executes the ensemble actions before generating the next chunk. In the real-world setup, the entire action chunk is executed at once. FALCON requires ~12.8 GB of GPU memory and runs at approximately 57 Hz on a single NVIDIA RTX 4090 GPU during real-world evaluation.

Table 7: **Hyper-parameters setup of FALCON for different experiments.** Abbreviations: Ep: Epochs, Iters: Iterations

| Experiment Name | Action Predictor | Window Size | Chunk Size | VLM Input View | ESM Input View | Batch Size | Learning Rate | Total |
|---|---|---|---|---|---|---|---|---|
| CALVIN Performance (Tab. 1) | LSTM | 16 | 10 | Side+Wrist | Side | 128 | 2e-5 (ABC→D) 5e-5 (ABCD→D) | 5 Ep |
| SimplerEnv Performance (Tab. 2-3) | MLP | 1 | 5 | Side | Side | 128 | 2e-5 | 150K Iters |
| Real-World Performance (Fig. 3-5, Fig. 7) | MLP | 1 | 5 | Side | Side | 512 | 2e-5 | 30 Ep |
| CALVIN Ablation (Tab. 4-5, Tab. 8, Tab. 10) | MLP | 1 | 10 | Side+Wrist | Side | 128 | 5e-5 (ABC→D) 2e-5 (ABCD→D) | 5 Ep |

## A.6 BENCHMARK DETAILS

**CALVIN** (Mees et al., 2022) is a simulation benchmark designed for evaluating long-horizon, language-conditioned robotic manipulation. It consists of four scene splits (A, B, C, and D), each representing a distinct environment configuration and featuring 24k human-teleoperated demonstrations annotated with language instruction. Each trajectory is less than 64-time steps, which includes 34 pre-defined basic skills: `rotate blue block right, move slider right, lift red block slider, place slider, turn off light bulb, turn off led light, push in drawer, lift blue block drawer, close drawer, lift pink block slider, lift pink block table, move slider left, open drawer, turn on light bulb, rotate blue block left, push blue block left, rotate red block right, turn on led light, push pink block right, push red block left, lift blue block table, place in drawer, rotate red block left, push pink block left, lift stacked blocks, lift blue block slider, push red block right.`

We train and test FALCON and VLA baselines on different training/test splits to fully analyze the capabilities. Standard evaluation protocols such as ABC→D and ABCD→D are employed to assess

the models' generalization capability to unseen environments and its robustness in long-horizon task compositions. During evaluation, the robot is required to complete a set of 5 consecutive tasks. The metrics are the success rates of finishing these sequential tasks and the average length of achieved tasks (*Avg. Len.*). All evaluations are implemented on D split, with 1000 rollouts and 5 consecutive sub-tasks for each rollout.

**SimplerEnv** (Li et al., 2024c) provides a benchmark for evaluating the transfer and generalization capabilities of models trained on large-scale real-world video data. It supports diverse manipulation setups across both the WidowX and Google Robot platforms, incorporating variations in lighting conditions, object textures, color distributions, and camera viewpoints. By faithfully replicating real-world conditions in a simulated environment, SimplerEnv enables reproducible and controlled evaluation of robot policies, facilitating rigorous benchmarking under settings that closely mirror private real-world systems such as Bridge V2 (Walke et al., 2023) and Google Robot (Brohan et al., 2022; 2023).

We adopt the following tasks in the WidowX + Bridge setting:

- **put the spoon on the towel.** In this setup, the spoon is positioned at one corner of a square on the tabletop, with the towel placed at a different corner. The square has sides measuring 15 cm in length. The orientation of the spoon alternates between horizontal and vertical, requiring the robot to adjust the orientation of its gripper accordingly. This configuration results in a total of 24 trials.

- **put carrot on plate.** This setup is similar to *put the spoon on the towel*, but the spoon is replaced with a carrot and the towel is substituted with a plate.

- **stack the green block on the yellow block.** In this experiment, a green block is positioned at one corner of a square on the tabletop, while a yellow block is placed at a different corner. Both blocks measure 3 cm in size. Two square configurations with 10 cm and 20 cm side lengths are used. This setup results in a total of 24 trials.

- **put eggplant into yellow basket.** An eggplant is positioned randomly within the right basin of a sink, while a yellow basket is placed in the left basin. The eggplant's placement varies in both location and orientation but is carefully arranged to remain easily graspable, avoiding proximity to the sink's edges. A total of 24 trials are conducted under this setup.

For the Google Robot setting, we test the following tasks:

- **pick coke can.** The task assigned to the robot is to pick up an empty Coke can from the table and lift it. Under the standard configuration, the environment is kept free of any distracting elements. The Coke can is arranged in three distinct positions: lying flat horizontally, lying flat vertically, and standing upright. For each of these positions, the can is placed at 25 specific grid points within a defined rectangular area on the table. This setup results in 25 experiments per position, totaling 75 trials across all orientations.

- **move {obj1} near {obj2}.** In the experiment, a set of three objects was arranged on the table in a triangular formation. For each trial, one object was assigned the role of the source, another was designated as the target, and the third served as a distractor. This setup resulted in six distinct trials for each triplet and triangular configuration. From a total of eight objects—blue plastic bottle, Pepsi can, orange, 7up can, apple, sponge, Coke can, and Redbull can—five triplets were randomly selected. Additionally, two triangular patterns, upright and inverted, were employed. This design produced a total of 60 trials.

- **(open/close) (top / middle / bottom) drawer.** In this setup, the robot is placed facing a cabinet equipped with three drawers and tasked with opening or closing a specific drawer. This experiment evaluates the robot's capability to handle articulated objects. The robot is positioned at nine distinct locations on a predefined grid within a rectangular area on the floor. With three drawers and two possible actions (opening or closing), the setup results in a total of 54 trials.

- **open top drawer; place apple into top drawer.** In this experiment, the robot is tasked with opening the top drawer and transferring an apple from the surface of the cabinet into the drawer. This setup evaluates the robot's ability to execute tasks that require

multiple sequential actions. The robot is positioned in three distinct locations on the floor, while the apple is placed at nine specific grid points on the cabinet surface, resulting in a total of 27 trials. At the start, the robot operates under the instruction to open the top drawer. Once the robot either signals task completion with a "terminate" token or reaches the midpoint of the allotted time, the instruction transitions to directing the robot to place the apple into the drawer.

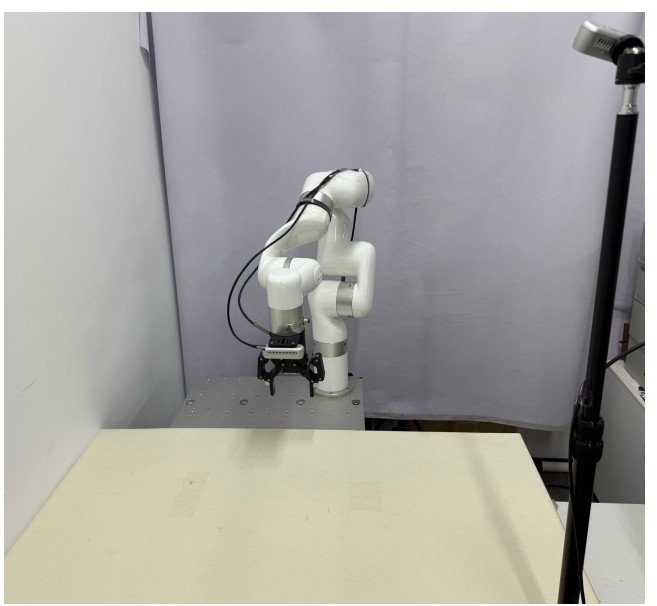
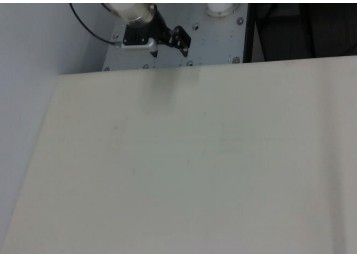

**Side Cam RGB**

**Side Cam Depth**

Figure 9: **Real-world setup of the xArm 6 robotic system used in the experiments.** The system is equipped with a side camera that provides both RGB and depth images for visual observation and spatial perception.

**Real World Benchmark** comprises 1,030 expert trajectories collected through human teleoperation, spanning five distinct robot learning scenarios and 11 individual tasks. These range from simple object interactions, such as *lift the yellow pepper*, to long-horizon sequential activities, such as *place the red coke can on the bottom shelf*, comprehensively assessing the model's robustness and spatial perception capabilities.

*Base Tasks* are organized into three scenarios (*Dining Table, Bedroom, and Kitchen*) containing a total of nine distinct task suites, encompassing language grounding (cluttered scenes with random distractors) and semantic understanding (unseen object poses). Each task is evaluated over 10 different scene layouts with 10 trials, resulting in a total of 90 rollouts. Besides, for each task, we collected 100 demonstration trajectories (except *lift the yellow pepper* for 50 trajectories), resulting in a total of 850 trajectories for training.

*Few-shot Adaptation* includes four challenging tasks selected from the *Base Tasks* that require more spatial perception capabilities. For each task, we collected 20 demonstration trajectories, resulting in a total of 80 trajectories for training. In addition to this base setting (denoted as *Simple* in Fig. 15), we introduce three unseen variations: *Unseen Object*, *Unseen Background* (by changing two different colored tablecloths), and *Unseen Task Description*, to evaluate the robustness and generalization of all models in low-data regimes. Each task is evaluated across 5 different layouts with 2 trials per layout.

*Spatial Understanding Capability Evaluations* consist of four tasks with varying levels of spatial complexity: two spatial-prompt tasks adapted via efficient fine-tuning (each task we collected 50 demonstrations), two **zero-shot** tasks, one from *Base Tasks* involving explicit height variation (*put white cup on pink cloth* with two 3cm blocks below cup), and the other from *Few-shot Adaptation* featuring objects of different sizes (*stack blue block on red block* with larger block size: 5cm, and smaller block size: 3cm. Regular size for training is 4cm). This suite of tasks is designed to further

investigate the spatial perception capabilities of the FALCON model. Each task is evaluated across 5 different layouts with 2 trials per layout.

The physical setup consists of an xArm 6 robot arm equipped with a Robotiq parallel gripper and an Intel RealSense D435i depth camera positioned approximately 0.6 meters away to provide a *third-person view*, as illustrated in Fig. 9. All fine-tuning datasets are collected via human teleoperation using a Spacemouse device, sampled at 10Hz. We use absolute Cartesian control as the action space for policy training and deployment (Liu et al., 2025b).

## A.7 ABLATION STUDY

### A.7.1 LOSS WEIGHTING FACTORS

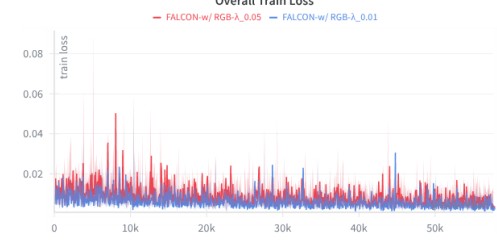

Table 8: **Performance comparison of different loss weighting factors on CALVIN benchmark.**

| Method | Task | Tasks Completed in a Row (%) | | | | | Avg. Len. ↑ |
|---|---|---|---|---|---|---|---|
| | | 1 | 2 | 3 | 4 | 5 | |
| FALCON ($\lambda = 0.05$) | ABC→D | 93.2 | 85.8 | 77.4 | 69.5 | 60.4 | 3.87 |
| **FALCON ($\lambda = 0.01$)** | ABC→D | **93.7** | **86.9** | **77.9** | **70.3** | **62.2** | **3.91** |

Figure 10: **Loss curves of FALCON under different weighting factors during training Stage 2 on CALVIN benchmark.**

For all experiments in our paper, we set $\lambda = 0.01$ for fair comparisons. To further ablate about the sensitivity of different $\lambda$, we conducted an additional experiment for FALCON with $\lambda = 0.05$ under RGB-only input on CALVIN zero-shot setting. As shown in Tab. 8, performance decreased slightly when $\lambda$ increased from 0.01 to 0.05 (*Avg. Len.* dropped from 3.91 to 3.87). Analysis of the overall train loss curves (Fig. 10) reveals that larger $\lambda$ leads to increased oscillation during training, while the overall convergence trend remains the same, which indicates that the model is not highly sensitive to small variations in $\lambda$.

### A.7.2 LSTM HEAD PARAMETERS

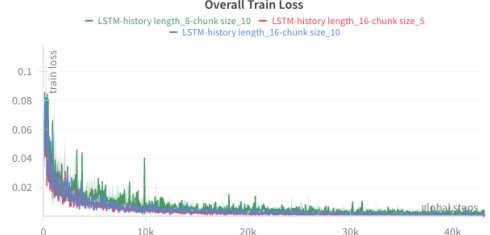

Table 9: **Performance comparison of Kosmos-VLA (LSTM) under different parameters on CALVIN benchmark.**

| Method | Task | Tasks Completed in a Row (%) | | | | | Avg. Len. ↑ |
|---|---|---|---|---|---|---|---|
| | | 1 | 2 | 3 | 4 | 5 | |
| **LSTM (*H=16, C=10*)** | ABC→D | **96.5** | **91.4** | **85.6** | **78.7** | **70.7** | **4.23** |
| LSTM (*H=16, C=5*) | ABC→D | 96.0 | 89.6 | 80.3 | 70.8 | 62.3 | 3.99 |
| LSTM (*H=8, C=10*) | ABC→D | 93.5 | 85.3 | 77.2 | 70.4 | 63.2 | 3.90 |

Figure 11: **Loss curves of Kosmos-VLA (LSTM) under different parameters on CALVIN benchmark.**

To systematically evaluate the effects of history length (H) and chunk size (C), we conducted experiments with Kosmos-VLA (RGB-only) using an LSTM action predictor under the CALVIN ABC→D setting.

**Performance Comparison.** We tested three configurations: (a) H=16, C=10, (b) H=16, C=5, and (c) H=8, C=10. The results shown in Tab. 9 demonstrate that H=16, C=10 achieves the strongest performance (*Avg. Len.* 4.23). Reducing either parameter degrades model performance. Specifically, a shorter history length (H=8) caused the most significant performance drop (*Avg. Len.* decreased from 4.23 to 3.90), underscoring that richer historical context is crucial for long-horizon tasks in robotic manipulation.

**Training Stability.** The overall train loss curves (Fig. 11) reveal that while all configurations eventually converge, reducing history length significantly impacts initial training stability. Specifically, H=8, C=10 shows slower descent and greater oscillation in early stages (0-15k steps), whereas reducing chunk size (C from 10 to 5) maintains stable convergence. This indicates that sufficient history length is vital not only for final performance but also for stable optimization.

## A.8 POTENTIAL FUTURE WORKS

Table 10: **Performance comparison of wrist camera input for ESM on the CALVIN benchmark.**

| Method | Task | Tasks Completed in a Row (%) | | | | | Avg. Len. ↑ |
|---|---|---|---|---|---|---|---|
| | | 1 | 2 | 3 | 4 | 5 | |
| w/o wrist | ABCD→D | 94.0 | 86.7 | 80.8 | 76.4 | **70.9** | 4.08 |
| **w/ wrist** | ABCD→D | **94.1** | **87.2** | **81.6** | **77.0** | 70.6 | **4.10** |

The integration of wrist camera images into the ESM further enhances FALCON's performance, as evidenced in Tab. 10. For instance, in the CALVIN ABCD→D setting, the *Avg. Len.* increases from 4.08 to 4.10 when wrist images are incorporated. This improvement suggests that multi-view inputs can provide complementary geometric cues. Future work could investigate multi-view camera systems that offer more consistent geometric perspectives, potentially further boosting robustness in diverse sensor configurations.

## A.9 ROLLOUT EXAMPLES IN CALVIN

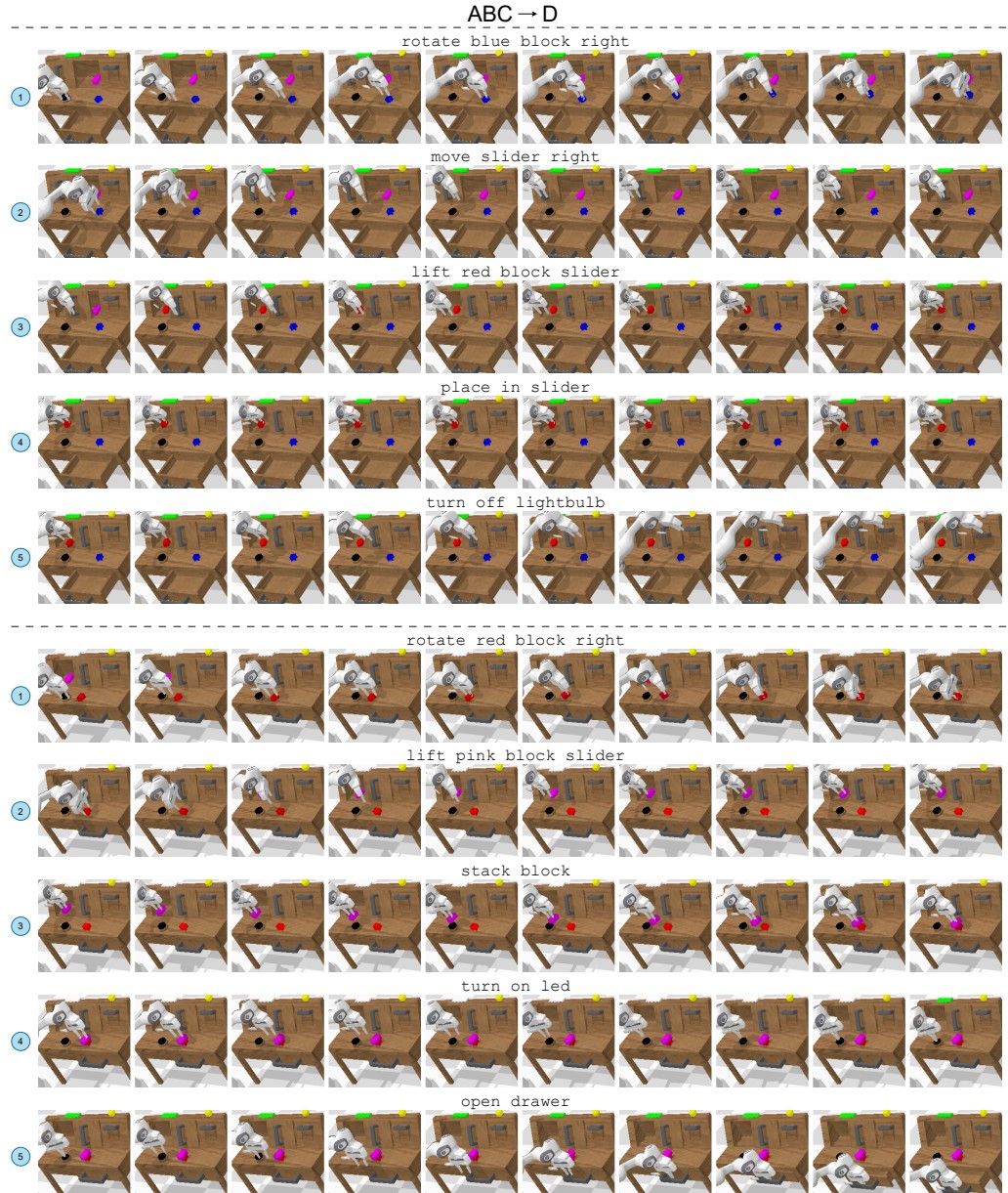

Figure 12: **Rollouts on the *ABC→D* split of the CALVIN benchmark.**

## A.10 ROLLOUT EXAMPLES IN SIMPLERENV

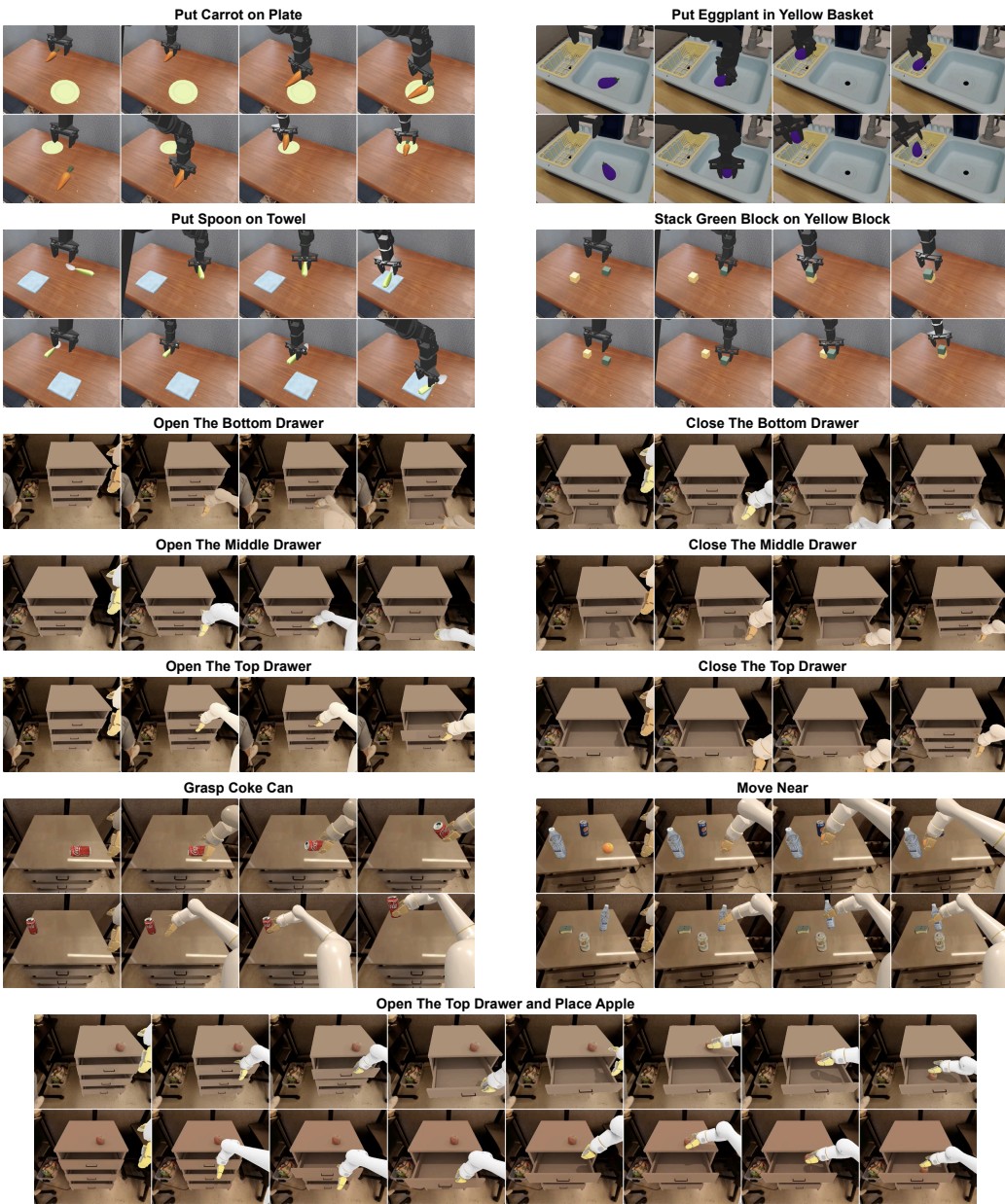

Figure 13: **Qualitative results for SimplerEnv tasks.**

## A.11 ROLLOUT EXAMPLES IN REAL-WORLD TASKS

### A.11.1 BASE TASKS ROLLOUTS

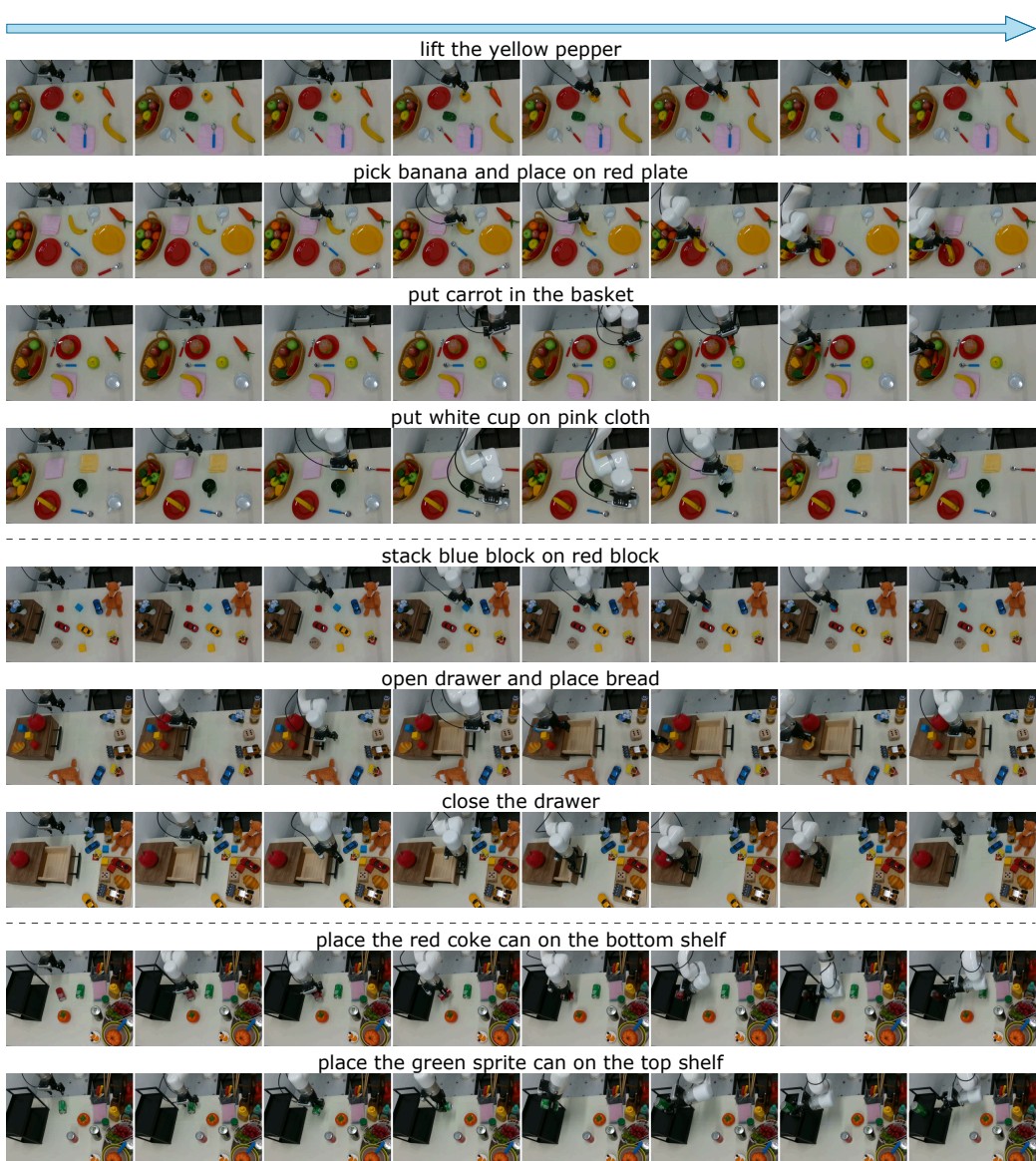

Figure 14: **Qualitative results for *Base Tasks*.**

### A.11.2 FEW-SHOT ADAPTATION ROLLOUTS AND DETAILED RESULTS

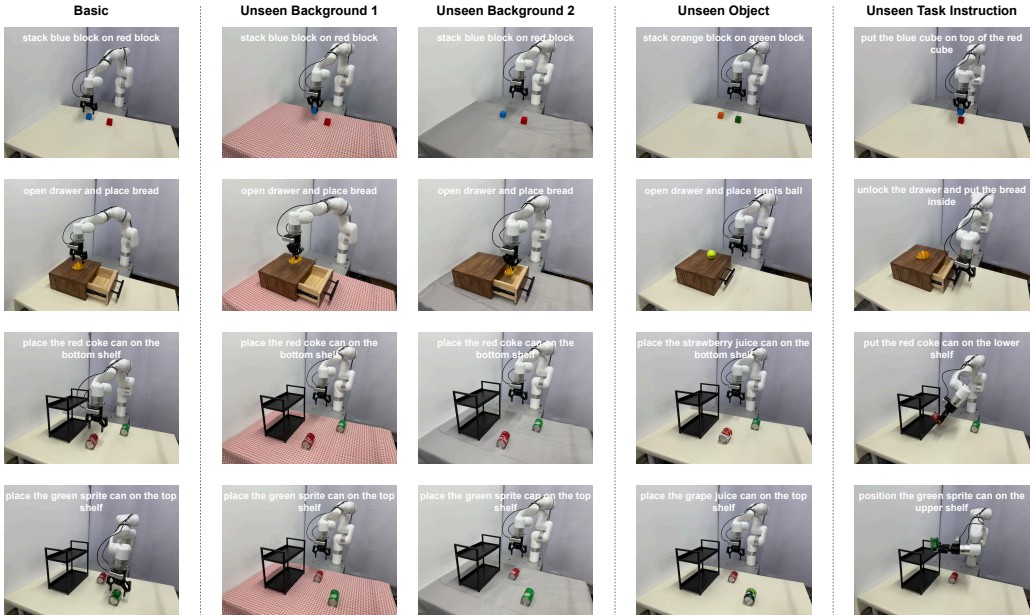

Figure 15: **Qualitative results for *Few-Shot Adaptation* tasks.**

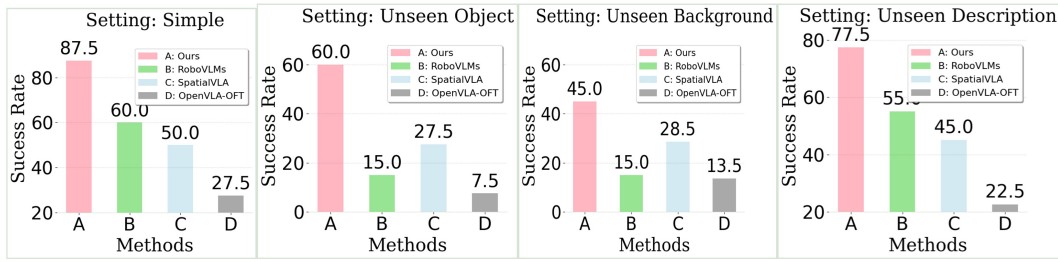

Figure 16: **Performance comparison of different methods** in the *Simple* setting and four variants.

Table 11: **Few-shot Adaptation performance under *Simple* Settings.** We report both the final success rates (Success) along with the sub-task success rates (e.g., Grasp Block). **Stack Block**: Stack Blue Block on Red Block. **Place Bread**: Open Drawer and Place Bread. **Place Coke Can**: Place the Red Coke Can on the Bottom Shelf. **Place Sprite Can**: Place the Green Sprite Can on the Top Shelf.

| Method | Stack Block | | Place Bread | | Place Coke Can | | Place Sprite Can | | Average |
|---|---|---|---|---|---|---|---|---|---|
| | Grasp Block | Success | Open Drawer | Success | Grasp Can | Success | Grasp Can | Success | |
| OpenVLA-OFT (Kim et al., 2025) | 30.0% | 30.0% | 30.0% | 30.0% | 40.0% | 30.0% | 60.0% | 20.0% | 27.5% |
| RoboVLM (Li et al., 2024a) | 60.0% | 40.0% | **100.0%** | 80.0% | **80.0%** | 60.0% | **100.0%** | 60.0% | 60.0% |
| SpatialVLA (Qu et al., 2025) | 60.0% | 50.0% | 90.0% | 70.0% | 50.0% | 40.0% | 80.0% | 50.0% | 50.0% |
| **FALCON (ours)** | **90.0%** | **80.0%** | **100.0%** | **100.0%** | **80.0%** | **80.0%** | 90.0% | **90.0%** | **87.5%** |

Table 12: **Few-shot Adaptation performance under *Unseen Objects* variants.** We report both the final success rates (Success) along with the sub-task success rates (e.g., Grasp Block). **Stack Block**: Stack Orange Block on Green Block. **Place Tennis Ball**: Open Drawer and Place Tennis Ball. **Place Strawberry Can**: Place the Strawberry Juice Can on the Bottom Shelf. **Place Grape Can**: Place the Grape Juice Can on the Top Shelf.

| Method | Stack Block | | Place Tennis Ball | | Place Strawberry Can | | Place Grape Can | | Average |
|---|---|---|---|---|---|---|---|---|---|
| | Grasp Block | Success | Open Drawer | Success | Grasp Can | Success | Grasp Can | Success | |
| OpenVLA-OFT (Kim et al., 2025) | 20.0% | 0.0% | 60.0% | 0.0% | 30.0% | 10.0% | 60.0% | 20.0% | 7.5% |
| RoboVLM (Li et al., 2024a) | **40.0%** | 0.0% | 60.0% | 0.0% | 60.0% | 20.0% | **80.0%** | 40.0% | 15.0% |
| SpatialVLA (Qu et al., 2025) | 30.0% | 10.0% | 70.0% | 20.0% | **70.0%** | 30.0% | 60.0% | 50.0% | 27.5% |
| **FALCON (ours)** | **40.0%** | **40.0%** | **100.0%** | **80.0%** | 60.0% | **60.0%** | 60.0% | **60.0%** | **60.0%** |

Table 13: **Few-shot Adaptation performance under *Unseen Background 1* variants.** We report both the final success rates (Success) along with the sub-task success rates (e.g., Grasp Block). **Stack Block**: Stack Blue Block on Red Block. **Place Bread**: Open Drawer and Place Bread. **Place Coke Can**: Place the Red Coke Can on the Bottom Shelf. **Place Sprite Can**: Place the Green Sprite Can on the Top Shelf.

| Method | Stack Block | | Place Bread | | Place Coke Can | | Place Sprite Can | | Average |
|---|---|---|---|---|---|---|---|---|---|
| | Grasp Block | Success | Open Drawer | Success | Grasp Can | Success | Grasp Can | Success | |
| OpenVLA-OFT (Kim et al., 2025) | 20.0% | 10.0% | 20.0% | 20.0% | 0.0% | 0.0% | 20.0% | 20.0% | 12.5% |
| RoboVLM (Li et al., 2024a) | 20.0% | 0.0% | 20.0% | 20.0% | 20.0% | 0.0% | 20.0% | 20.0% | 10.0% |
| SpatialVLA (Qu et al., 2025) | 20.0% | 20.0% | 40.0% | 30.0% | 20.0% | 20.0% | 30.0% | **30.0%** | 25.0% |
| **FALCON (ours)** | **40.0%** | **40.0%** | **60.0%** | **60.0%** | **40.0%** | **40.0%** | **40.0%** | 20.0% | **40.0%** |

Table 14: **Few-shot Adaptation performance under *Unseen Background 2* variants.** We report both the final success rates (Success) along with the sub-task success rates (e.g., Grasp Block). **Stack Block**: Stack Blue Block on Red Block. **Place Bread**: Open Drawer and Place Bread. **Place Coke Can**: Place the Red Coke Can on the Bottom Shelf. **Place Sprite Can**: Place the Green Sprite Can on the Top Shelf.

| Method | Stack Block | | Place Bread | | Place Coke Can | | Place Sprite Can | | Average |
|---|---|---|---|---|---|---|---|---|---|
| | Grasp Block | Success | Open Drawer | Success | Grasp Can | Success | Grasp Can | Success | |
| OpenVLA-OFT (Kim et al., 2025) | **40.0%** | 10.0% | 20.0% | 20.0% | 20.0% | 10.0% | 40.0% | 20.0% | 15.0% |
| RoboVLM (Li et al., 2024a) | **40.0%** | 0.0% | 20.0% | 20.0% | 20.0% | 20.0% | **60.0%** | **40.0%** | 20.0% |
| SpatialVLA (Qu et al., 2025) | 30.0% | 30.0% | 40.0% | 30.0% | 50.0% | 40.0% | 50.0% | 30.0% | 32.5% |
| **FALCON (ours)** | **40.0%** | **40.0%** | **60.0%** | **60.0%** | **60.0%** | **60.0%** | **60.0%** | **40.0%** | **50.0%** |

Table 15: **Few-shot Adaptation performance under *Unseen Task Description* variants.** We report both the final success rates (Success) along with the sub-task success rates (e.g., Put Cube). **Put Cube**: Put the Blue Cube on Top of the Red Cube. **Put Bread**: Unlock the Drawer and Put the Bread Inside. **Put Coke Can**: Put the Red Coke Can on the Lower Shelf. **Position Sprite Can**: Position the Green Sprite Can on the Upper Shelf.

| Method | Put Cube | | Put Bread | | Put Coke Can | | Position Sprite Can | | Average |
|---|---|---|---|---|---|---|---|---|---|
| | Grasp Block | Success | Open Drawer | Success | Grasp Can | Success | Grasp Can | Success | |
| OpenVLA-OFT (Kim et al., 2025) | 40.0% | 40.0% | 20.0% | 20.0% | 40.0% | 20.0% | 40.0% | 10.0% | 22.5% |
| RoboVLM (Li et al., 2024a) | 60.0% | 40.0% | **100.0%** | 60.0% | **100.0%** | **80.0%** | **100.0%** | 40.0% | 55.0% |
| SpatialVLA (Qu et al., 2025) | 50.0% | 40.0% | 80.0% | 60.0% | 40.0% | 40.0% | 70.0% | 40.0% | 45.0% |
| **FALCON (ours)** | **70.0%** | **70.0%** | **100.0%** | **100.0%** | 80.0% | **80.0%** | 80.0% | **60.0%** | **77.5%** |

### A.11.3 SPATIAL UNDERSTANDING CAPABILITY EVALUATION ROLLOUTS

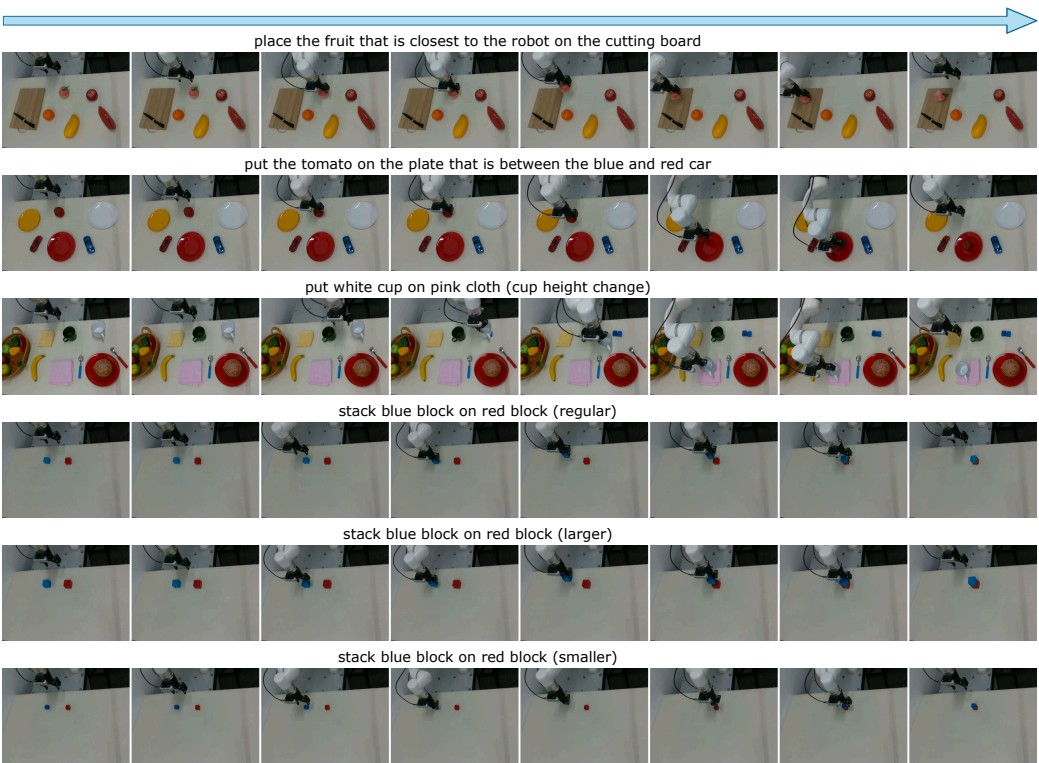

Figure 17: **Qualitative results for *Spatial Understanding Capability Evaluations*.**

