# OpenReview forum: "From Spatial to Actions: Grounding Vision-Language-Action Model in Spatial Foundation Priors"
_ICLR.cc/2026/Conference — ICLR 2026 Poster_

### Official Review · Reviewer_jhgj · 2025-10-31

**Soundness:** 4
**Presentation:** 4
**Contribution:** 4
**Rating:** 8
**Confidence:** 4

**Summary:**

The paper proposes FALCON, a Vision-Language-Action (VLA) architecture designed to enhance 3D spatial reasoning by injecting rich 3D spatial tokens—derived from a Spatial Foundation Model—into a dedicated "Spatial-Enhanced Action Head," rather than forcing them through the VLM (Vision Language Model) backbone. It introduces an "Embodied Spatial Model" (ESM) that (i) can operate RGB-only via VGGT-like spatial priors and (ii) optionally fuses depth and pose information without retraining or architectural changes, using a stochastic 3D-conditioning scheme to preserve robustness across modalities. Fusion strategies between the semantic action token and the spatial token were studied (cross-attention, FiLM-gated, element-wise addition), with element-wise addition selected for stability and generalization. Extensive experiments across CALVIN, SimplerEnv (WidowX/Google Robot), and 11 real-world tasks show that FALCON achieves SOTA or highly competitive results. Ablations demonstrate its modality transferability and support the claim that injecting tokens into the action head helps preserve VLM alignment.

**Strengths:**

1. Clear problem framing: Identifies the 2D–3D gap in VLA spatial reasoning and the modality transferability and alignment issues with prior explicit 3D pipelines and weak 3D cues.

2. Architectural novelty at the integration point: Instead of injecting spatial tokens into the VLM (which can disturb alignment), FALCON routes rich spatial tokens to a Spatial-Enhanced Action Head. This design decision is well-motivated and experimentally supported by a key ablation.

3. Practical modality transferability: RGB-only works via spatial foundation priors; optional depth/pose yields measurable gains, with a principled stochastic-conditioning scheme during training.

4. Extensive experiments: Strong results on CALVIN and SimplerEnv, clear real-world evaluation across 11 tasks, and targeted spatial capability tests (scale/height/abstract spatial prompts). Ablations on fusion strategies and token injection positions are informative.

5. Reproducibility-minded details: Training stages, losses, data splits, and hyperparameters are detailed; the ESM depth quality is quantitatively analyzed.

**Weaknesses:**

1. Positioning vs. contemporaneous 3D-enhanced VLAs: The related work section is broad, but could better differentiate FALCON from the latest VLA methods employing similar spatial decoupling strategies (e.g., avoiding direct modification of the VLM backbone). Clearer articulation of FALCON's specific innovations compared to potentially similar works and a systematic comparison of training costs (e.g., compute requirements) and data dependencies are recommended.

2. Theoretical grounding of fusion choice: The paper empirically finds "element-wise addition" to be the optimal fusion strategy but lacks theoretical support explaining this counter-intuitive result. Why does this non-parametric method outperform more complex parametric fusions (like Cross-Attention)? Further analysis, for instance, by quantifying the alignment drift (Δ) in the VLM's representation space under different fusion strategies, could provide deeper insight into why element-wise addition better preserves generalization.

3. Quantifying robustness: The proposed stochastic conditioning training strategy is an interesting design for enhancing modal robustness. However, the paper lacks a quantitative analysis of this robustness. It is recommended to add experiments that formally evaluate model performance under missing or noisy modal information (e.g., varying ratios r of valid depth pixels), potentially by plotting performance degradation curves.

4. Sensitivity to spatial prior source: FALCON's ESM module draws inspiration from VGGT. Given the existence of other spatial foundation models (like DUSt3R and MASt3R, cited in the paper), how sensitive is FALCON's performance to the specific choice of spatial prior model? Ablation studies, such as replacing the ESM backbone with DUSt3R or MASt3R, are suggested to test the generality of the FALCON architecture and assess potential domain adaptation issues.

5. Data efficiency substantiation: When evaluating few-shot generalization, the paper primarily reports final performance based on a fixed number (e.g., 20) of demonstrations. To provide a more comprehensive view of data efficiency, presenting learning curves—showing how success rate varies with the number of training demonstrations (e.g., 1, 5, 10, 20 demos)—is recommended to better illustrate the model's ability to learn from small samples.

**Questions:**

1. Could you quantify the representation drift in the VLM when spatial tokens are injected into the VLM vs. the action head (e.g., using CKA or feature-space distance to compare text-image alignment before/after injection)?

2. How sensitive are results to the Bernoulli probabilities p for depth/pose injection? Please provide a sweep and show their effect on RGB-only inference performance.

3. Can you provide success vs. number of real-world demonstrations curves for at least two tasks to substantiate few-shot efficiency?

4. Have you tried swapping VGGT with DUSt3R/MASt3R/CUT3R as the ESM backbone, holding everything else fixed? Was there any change in performance or modality transferability?

5. For long-horizon tasks (like CALVIN), what is the effect of the LSTM history length H and chunk size C on stability and success rate?

6. What is the runtime overhead (e.g., latency or FPS) of enabling the depth/pose inputs for the ESM compared to RGB-only inference on a single GPU?

---

> ### Author Response · Authors · 2025-11-24
> **Rebuttal 1: W1**
>
> We sincerely thanks for your critical feedback and positive assessment of our work. We are particularly grateful for your recognition of our “clear problem framing”, “novel architectural design”, and "strong experimental validation". Besides, the concerns and questions you pointed out have provided valuable insights and we respond to each of them point by point below. In the revised manuscript, we mark our major revisions as $\textbf{\textcolor{blue}{blue}}$ for your easy reference.
>
> > 📝 W1: Positioning vs. contemporaneous 3D-enhanced VLAs: The related work section is broad, but could better differentiate FALCON from the latest VLA methods employing similar spatial decoupling strategies (e.g., avoiding direct modification of the VLM backbone). Clearer articulation of FALCON's specific innovations compared to potentially similar works and a systematic comparison of training costs (e.g., compute requirements) and data dependencies are recommended.
>
> 💡 **A:** Thank you for this constructive suggestion. We have revised the **Related Work section (See Appendix A.2.1)** to more clearly articulate FALCON’s novelties compared to the latest VLA methods employing similar spatial decoupling strategies: PointVLA (2025) [a] and GeoVLA (2025) [b]. Specifically:
>
> 1. Unlike methods like GeoVLA and PointVLA that rely on explicit 3D inputs (e.g., point clouds), FALCON achieves superior modality transferability, enabling flexible deployment in diverse real-world settings. Besides, Our additinal experiments confirm that **FALCON can gain performance at test-time by incorporating depth when available, even when trained solely on RGB (see L479-485 and Table 5 at Sec. 3.3 for more detailed analysis)**.
> 2. To provide a systematic comparison, we summarize key characteristics at **Table D1**. As illustrated, FALCON’s ability to **operate effectively with RGB-only input while optionally exploiting additional geometric inputs without retraining or architecture modification** distinguishes it from contemporaneous approaches. This enables effective utilization of large-scale datasets like Open X-Embodiment (OXE) [c] which lack precise 3D annotations and reduces dependency on specialized sensor setups in real-world deployment.
>
> **Table D1: Comparison of 3D-enhanced VLAs across key characteristics.**
> | Method | Model Size | CALVIN Support | OXE Support | Real-World Deployment | Modality Transferability |
> |--------|------------|----------------|-------------|----------------------|----------------------|
> | FALCON | ~2.9B | ✓ | ✓ | ✓ | High (RGB-only + optional depth/pose) |
> | GeoVLA | ~7.7B¹ | ✓ | ✗² | Challenging³ | Low (requires point clouds) |
> | PointVLA | ~3.1B¹ | ✓ | ✗² | Challenging³ | Low (requires point clouds) |
>
> *Notes:
> ¹ Model sizes for GeoVLA and PointVLA are estimated as neither paper provided exact model sizes and their code is not open-sourced. Direct comparison of training costs is unfair and infeasible due to different datasets/GPUs used.
> ² OXE dataset lacks aligned 3D annotations, limiting methods dependent on explicit 3D inputs.
> ³ Real-world deployment requires additional sensor calibration and point cloud projection.*
>
> [a] PointVLA: Injecting the 3D World into Vision-Language-Action Models
>
> [b] GeoVLA: Empowering 3D Representations in Vision-Language-Action Models
>
> [c] Open X-Embodiment: Robotic Learning Datasets and RT-X Models

---

> ### Author Response · Authors · 2025-11-24
> **Rebuttal 2: W2, W3**
>
> > 📝 W2: Theoretical grounding of fusion choice: The paper empirically finds "element-wise addition" to be the optimal fusion strategy but lacks theoretical support explaining this counter-intuitive result. Why does this non-parametric method outperform more complex parametric fusions (like Cross-Attention)? Further analysis, for instance, by quantifying the alignment drift (Δ) in the VLM's representation space under different fusion strategies, could provide deeper insight into why element-wise addition better preserves generalization.
>
> 💡 **A:** Thank you for this insightful question regarding the theoretical justification for our fusion strategy selection. To quantitatively analyze the alignment drift (Δ) in the VLM's representation space under different fusion strategies, we conducted a carefully designed experiment on the CALVIN benchmark evaluation set.
>
> 1. **Experimental setup:** We extracted image-text tokens from VLM's last hidden state outputs of different fusion variants and computed the **L2 distance** against the corresponding tokens from the baseline Kosmos-VLA (without spatial token injection). This distance serves as our quantitative measure of alignment drift (Δ) inside the VLM.
> 2. **Analysis:** As shown in **Table D2**, the results reveal a clear **negative correlation between alignment drift and generalization performance on CALVIN zero-shot setting**. Specifically, **element-wise addition achieves the lowest alignment drift (10.05) while delivering the best generalization performance (Avg. Len. 3.91)**. This supports our findings on CALVIN presented in **Table 4** that straightforward and non-parametric fusion strategies better preserve VLM's semantic alignment, which is crucial for zero-shot generalization.
> 3. **Theoretical interpretation:** From a theoretical perspective, unlike parametric methods (e.g., cross-attention) that learn new transformations potentially causing feature distortion, or gated mechanisms that may selectively suppress important semantic features, addition preserves the original feature directions while incorporating spatial information as a complementary signal.
>
> **Table D2:** **Alignment drift and generalization performance on CALVIN benchmark across different fusion strategies.**
> | Fusion Method | Avg. Alignment Drift (Δ) ↓ | CALVIN ABC→D Avg. Len. ↑ |
> |---------------|----------------------------------|---------------------------|
> | **Element-wise Addition** | **10.05** | **3.91** |
> | FiLM-Gated Modulation | 10.21 | 3.76 |
> | Cross-Attention | 10.96 | 3.68 |
>
> > 📝 W3: Quantifying robustness: The proposed stochastic conditioning training strategy is an interesting design for enhancing modal robustness. However, the paper lacks a quantitative analysis of this robustness. It is recommended to add experiments that formally evaluate model performance under missing or noisy modal information (e.g., varying ratios r of valid depth pixels), potentially by plotting performance degradation curves.
>
> 💡 **A:** Thanks for the valuable comments. We conducted an ablation study **by adding Gaussian noise to the injected depth condition during zero-shot monocular depth estimation on CALVIN benchmark**. The results are shown in **Figure 13 at Appendix A.7.4**. When adding 10\%–30\% noise, the ESM performance with additinal depth inputs drops from 99.79\% to 77.96\%, and the performance without depth inputs mantains the same. This indicates that injecting noise into the depth condition during test-time makes the depth prediction task more challenging and further degrades the overall performance of the network.

---

> ### Author Response · Authors · 2025-11-24
> **Rebuttal 3: W4, Q4**
>
> > 📝 W4: Sensitivity to spatial prior source: FALCON's ESM module draws inspiration from VGGT. Given the existence of other spatial foundation models (like DUSt3R and MASt3R, cited in the paper), how sensitive is FALCON's performance to the specific choice of spatial prior model? Ablation studies, such as replacing the ESM backbone with DUSt3R or MASt3R, are suggested to test the generality of the FALCON architecture and assess potential domain adaptation issues.
>
> > 📝 Q4: Have you tried swapping VGGT with DUSt3R/MASt3R/CUT3R as the ESM backbone, holding everything else fixed? Was there any change in performance or modality transferability?
>
> 💡 **A:** Thank you for this valuable question regarding the sensitivity of FALCON to the choice of spatial foundation model. We sincerely apologize that due to resource and time constraints, we were unable to perform an exhaustive ablation with all mentioned models (DUSt3R, MASt3R, CUT3R) and swapping VGGT with them then retrain the Embodied Spatial Model (ESM). However, we did conduct an additional experiment by replacing the whole ESM module with DUSt3R [d] while keeping all other components fixed.
>
> **We substituted the spatial tokens from ESM with the Transformer Decoder output tokens from DUSt3R** and evaluated it on the CALVIN ABC→D setting. As shown in **Table D4**, **while FALCON with DUSt3R still outperforms the 2D baseline (Avg. Len. increased from 3.48 to 3.70), it significantly underperforms our ESM-based model (3.70 vs. 3.91).** We believe this performance gap stems from the **quality of spatial priors**. To verify the geometric reasoning capability of each module, we evaluated their monocular depth estimation performance on the Sintel [e] dataset (RGB-only input). The results in **Table D5** confirm that our ESM provides superior geometric priors, thus it's reseanable to deliver a better results on the CALVIN zero-shot setting.
>
> **Modality transferability:** A key architectural limitation of DUSt3R is its inability to utilize and fuse additional geometric inputs like depth or camera pose. When replacing ESM with DUSt3R, FALCON loses its crucial modality transferability, becoming a purely RGB-based model and thus limiting its peak performance potential when auxiliary information are available.
>
> In summary, FALCON's performance will be affected by the quality and capabilities of the spatial prior source. Our proposed ESM, with its strong geometric priors and native support for multi-modal inputs, is critical for FALCON achieving SOTA performance and robust modality transferability. Therefore, although FALCON's architecture is general, the choice of a strong and flexible spatial module is important.
>
> **Table D4:** **Performance comparison of different spatial module for FALCON on CALVIN benchmark.**
> | Method | Task | SR1 | SR2 | SR3 | SR4 | SR5 | Avg. Len. ↑ |
> |--------|------|---|---|---|---|---|-------------|
> | Kosmos-VLA (w/ rgb) | ABC→D | 91.6% | 80.0% | 68.3% | 58.8% | 49.0% | 3.48 |
> | FALCON (DUSt3R) | ABC→D | 91.5% | 82.8% | 73.9% | 65.1% | 56.6% | 3.70 |
> | **FALCON (ESM)** | ABC→D | **93.7%** | **86.9%** | **77.9%** | **70.3%** | **62.2%** | **3.91** |
>
> **Table D5:** **Monocular depth evaluation on Sintel dataset.**
> | Method (w/ rgb) | δ < 1.25 (%) ↑ | Abs. Rel ↓ |
> |---------------|----------------------------------|---------------------------|
> | DUSt3R  | 58.7 | 0.424 |
> | **ESM** | **67.5** | **0.273** |
>
> [d] DUSt3R: Geometric 3D Vision Made Easy
>
> [e] A Naturalistic Open Source Movie for Optical Flow Evaluation

---

> ### Author Response · Authors · 2025-11-24
> **Rebuttal 4: W5, Q3, Q1**
>
> > 📝 W5: Data efficiency substantiation: When evaluating few-shot generalization, the paper primarily reports final performance based on a fixed number (e.g., 20) of demonstrations. To provide a more comprehensive view of data efficiency, presenting learning curves—showing how success rate varies with the number of training demonstrations (e.g., 1, 5, 10, 20 demos)—is recommended to better illustrate the model's ability to learn from small samples.
>
> > 📝 Q3: Can you provide success vs. number of real-world demonstrations curves for at least two tasks to substantiate few-shot efficiency?
>
> 💡 **A:** Thank you for this valuable feedback. Following your suggestion, we have conducted additional experiments to provide a more comprehensive analysis of FALCON's few-shot efficiency. We evaluated two few-shot adaptation tasks: (a) *Stack Blue Block on Red Block* and (b) *Open Drawer and Place Bread*, under varying numbers of training demonstrations (1, 5, 10, 20), while keeping all other conditions identical and using only RGB inputs. The baseline model is RoboVLM [f] and learning curves are provided in **Figure 12 at Appendix A.7.3**, which clearly demonstrate **FALCON's superior data efficiency with consistent advantage across all data regimes**.
>
> In the **extremely low-data regime (only 1 demo)**, FALCON achieves 10%-20% success rates while RoboVLM fails completely (0%) on the stacking task. As demonstration numbers increase, **FALCON maintains a consistent performance advantage**, achieving a twofold improvement in the success rate (80%) than RoboVLM (40%) with 20 demos on the stacking task, which substantiates FALCON's integration of geometric priors significantly enhances its ability to learn from limited samples. Moreover, for the sequential task (*open drawer and place bread*), **FALCON reaches perfect performance (100%) with just 20 demos, while RoboVLM plateaus at 80%**, highlighting our method's superior data efficiency in learning complex manipulation task.
>
> > 📝 Q1: Could you quantify the representation drift in the VLM when spatial tokens are injected into the VLM vs. the action head (e.g., using CKA or feature-space distance to compare text-image alignment before/after injection)?
>
> 💡 **A:** Thank you for raising this important question. To quantitatively compare the text-image alignment drift when injecting spatial tokens into the Action Head/VLM, we extended the experimental setup from the above W2.
>
> 1. **Experimental setup:** We evaluated both injection paradigms on the CALVIN benchmark evaluation set. For each paradigm, we extracted the VLM's output image-text tokens from last hidden states (after spatial token injection) and computed their L2 distance against the corresponding tokens from the Kosmos-VLA (before injection), which directly quantifies the disruption to the VLM's semantic space.
> 2. **Analysis:** As shown in **Table D3**, injecting spatial tokens directly into the Action Head causes lower alignment drift (19.70) in the VLM compared to VLM injection (20.15). This preservation of the semantic space directly translates to superior zero-shot performance, as evidenced by the higher Avg. Len. (3.91 vs. 3.79). The results quantitatively validate our core design principle: integrating spatial information into the Action Head effectively enhances geometric reasoning while maintaining the VLM's crucial vision-language alignment.
>
> **Table D3:** **Alignment drift and generalization performance on CALVIN benchmark across different injection positions.**
> | Injection Position | Avg. Alignment Drift (Δ) ↓ | CALVIN ABC→D Avg. Len. ↑ |
> |---------------|----------------------------------|---------------------------|
> | **Action head injection** | **19.70** | **3.91** |
> | VLM injection | 20.15 | 3.79 |
>
> [f] Towards Generalist Robot Policies: What Matters in Building Vision-Language-Action Models

---

> ### Author Response · Authors · 2025-11-24
> **Rebuttal 5: Q2, Q5, Q6**
>
> > 📝 Q2: How sensitive are results to the Bernoulli probabilities p for depth/pose injection? Please provide a sweep and show their effect on RGB-only inference performance.
>
> 💡 **A:** Thanks for your valuable question. The $p$ in $b_{\text{d}}, b_{\text{p}} \sim \text{Bernoulli}(p)$ is a training hyperparameter that specifies the probability of injecting depth/pose conditions during ESM training. This stochastic conditioning strategy enables robust scene reconstruction under both RGB-only and multi-modal inputs at inference time.
>
> Following your suggestion, we conducted an ablation study by sweeping $p$ across 50\%, 66\% (our standard setting), 80\%, 90\% and evaluating zero-shot monocular depth estimation performance on CALVIN. As shown in **Figure 14 at Appendix A.7.5**, $p = 66\\%$ yields the best performance trade-off between the with-depth and without-depth configurations. Moreover, under RGB-only inference, performance shows only a slight decrease (from 91.02\% to 87.48\%) as $p$ increases from 50\% to 90\%, demonstrating a **clear but modest negative correlation**. These results indicate that while higher $p$ values slightly increase depth-conditioned performance, **ESM's RGB-only inference remains relatively stable across different $p$ values**.
>
> > 📝 Q5: For long-horizon tasks (like CALVIN), what is the effect of the LSTM history length H and chunk size C on stability and success rate?
>
> 💡 **A:** Thank you for this insightful question. To systematically evaluate the effects of history length (H) and chunk size (C), we conducted experiments with Kosmos-VLA (RGB-only) using an LSTM action predictor under the CALVIN ABC→D setting. **Related loss curves are available at Appendix A.7.2.**
>
> 1. **Performance comparison:** We tested three configurations: (a) H=16, C=10, (b) H=16, C=5, and (c) H=8, C=10. The results in **Table D6** demonstrate that H=16, C=10 achieves the strongest performance **(Avg. Len. 4.23)**. **Reducing either parameter degrades model performance**. Specifically, a shorter history length (H=8) caused the most significant performance drop (Avg. Len. decreased from 4.23 to 3.90), underscoring that **richer historical context is crucial for long-horizon tasks in robotic manipulation**.
> 2. **Training stability:** The overall train loss curves **(Fig. 11 in Appendix A.7.2)** reveal that **while all configurations eventually converge, reducing history length significantly impacts initial training stability**. Specifically, H=8, C=10 shows **slower descent and greater oscillation** in early stages **(0-15k steps)**, whereas reducing chunk size (C from 10 to 5) maintains stable convergence. This indicates that **sufficient history length is vital not only for final performance but also for stable optimization**.
>
> **Table D6:** **Performance comparison of Kosmos-VLA (LSTM) under different parameters on CALVIN benchmark.**
> | Method | Task | SR1 | SR2 | SR3 | SR4 | SR5 | Avg. Len. ↑ |
> |--------|------|---|---|---|---|---|-------------|
> | **LSTM (H=16, C=10)** | ABC→D | **96.5%** | **91.4%** | **85.6%** | **78.7%** | **70.7%** | **4.23** |
> | LSTM (H=16, C=5) | ABC→D | 96.0% | 89.6% | 80.3% | 70.8% | 62.3% | 3.99 |
> | LSTM (H=8, C=10) | ABC→D | 93.5% | 85.3% | 77.2% | 70.4% | 63.2% | 3.90 |
>
> > 📝 Q6: What is the runtime overhead (e.g., latency or FPS) of enabling the depth/pose inputs for the ESM compared to RGB-only inference on a single GPU?
>
> 💡 **A:** Thank you for raising this important practical consideration. To quantify the runtime overhead of enabling additional input modalities, we conducted latency measurements for FALCON on a single NVIDIA RTX 4090 GPU using a third-view image inference for generating a single action.
>
> We measured the inference speed under two configurations and got the following results:
> * RGB-only: FALCON runs at approximately 57 Hz
> * With depth/pose inputs: FALCON maintains 53.6 Hz
>
> The introduction of auxiliary inputs caused only a **marginal latency increase (∼6% relative decrease in frequency)**. Crucially, GPU memory consumption remains identical at ∼12.8 GB in both configurations. These results confirm that **FALCON maintains real-time capability while leveraging additional spatial signals**, making it suitable for practical deployment where sensor availability may vary without compromising overall system performance.

---

### Official Review · Reviewer_Byfb · 2025-10-31

**Soundness:** 3
**Presentation:** 2
**Contribution:** 3
**Rating:** 4
**Confidence:** 4

**Summary:**

FALCON is a VLA that also accepts optional depth and camera pose inputs. These additional modalities can help with fine-grained manipulation tasks.  The approach will benefit from continued improvement in pointmap and camera pose estimation, and that technology is itself rapidly improving beginning with Dust3r).

The proposed approach is to start with an off-the-shelf RGB + text VLA, Kosmos, and add a large downstream transformer that accepts the Kosmos original output plus optional additional depth and camera pose modalities. The majority of the paper discusses the design decisions around this second network, which the authors call the "Embodied Spatial Module" (ESM). Many of the design decisions build on VGGT (not worried about novelty; stating this to be clear).

Results on CALVIN and SimpleBench allow comparisons to other VLAs, and results look strong. There are also some experiments on a real Franka.

Overall I feel the paper makes a useful contribution to the community. The empirical results are convincing but there are some small claims that do not feel fully substantiated and the paper could use some polish. With presentational changes I could be convinced to raise my score.

**Strengths:**

### Overall
Overall I feel the paper makes a useful empirical contribution -- especially if weights and code are released.

* Strong empirical results
* Straightforward method (this is a good thing!) with clear use case
* Clear description of the approach
* Good ablations of different design decisions
* Clear figures

Please don't take the length of this section an indicator. Mainly, these are the strengths that I'm seeing -- hopefully these are the same things the authors see too.

**Weaknesses:**

### Presentation
I found the paper a bit hard to read in its current form, and I feel the paper's impact may be limited by the writing and presentation.  Many parts of the paper could be tightened through removing unnecessary adjectives and reorganizing sections.

**Organization:**
In the current version, even basic information about the training approach, including the losses, learning algorithm, and datasets are left entirely to the appendix. In contrast, the method section focuses almost exclusively on the architecture and has multiple paragraphs devoted to equations ablations the authors will run (e.g. Sec 2.4).

In my opinion, this section could be reduced to equation 5 plus a little context, and the ablations could be moved to the experiments or appendix. This would make room for information about the training of FALCON.

**Wording**
Parts of the main paper are especially wordy and flowery.  Phrases like "Seamlessly integrating" are very common from LLMs but are hard to evaluate technically -- used three times in the paper. Wording like this could be cut to make space for some of the interesting ablations and details from the supplementary. I actually preferred the more direct writing in the supplementary!

---
### Experiments

**Cross-modality generalization**
There are lots of claims and experiments about cross-modality generalization, but I didn't find the experiments especially convincing and I don't feel these claims add much to the paper in their current form. Or maybe the authors could clarify the definition of "modality transferability" in the paper.

I would like modality transferability experiments showing that a task could be learned (ABCD->D) entirely without depth, and then benefit from depth at test-time (and/or be learned with depth, and not suffer too much from losing depth at test-time) -- these would be more convincing than the current table 5.

However, I wanted to check my understanding about table 5, which the authors say L430-431
> "ESM achieves performance comparable to VGGT when using only RGB input. Moreover, its performance improves significantly when additional depth or camera pose information is accessible. This demonstrates the inherent strength of FALCON’s modality transferability"

VGGT is a pointmap estimation model (no actions), and based on the metrics, it looks like this is performance for depth estimation? So the ESM's depth estimation on CALVIN (which the model was trained on), is similar to VGGT (which VGGT wasn't trained on). And then when the GT depth is passed to the ESM, the depth estimation is close to perfect?

**Questions:**

* How come the authors add together imd + depth tokens (equation 4) and use this strategy, instead of concatenating and either dropping tokens or masking them (like in flow matching/diffusion)?
* When training and evaluating FALCON, do the authors use the depth and camera poses from the simulator/Realsense + IK? Or is VGGT used at test-time?

---

> ### Author Response · Authors · 2025-11-24
> **Rebuttal 1: W1**
>
> We sincerely thank you for your thoughtful and constructive feedback, as well as for your recognition of the empirical contributions and clarity of our method. We are greatly encouraged by your positive assessment regarding the straightforwardness of our approach, the strong results, and the insightful ablation studies. Below, we thoroughly address each of your comments. All revised sections in the manuscript are highlighted in $\textbf{\textcolor{blue}{blue}}$ for your easy reference.
>
> ### **Overall Contribution and Code Release**
>
> > 📝 "Overall I feel the paper makes a useful empirical contribution -- especially if weights and code are released."
>
> 💡 **A:** Thank you for this positive assessment! We will release the code and relevant model checkpoints upon acceptance. We hope this will facilitate further research in VLA models and robotic manipulation.
>
> ### **1. Presentation and Organization**
>
> > 📝 W1: I found the paper a bit hard to read in its current form, and I feel the paper's impact may be limited by the writing and presentation. Many parts of the paper could be tightened through removing unnecessary adjectives and reorganizing sections.
>
> 💡 **A:** We sincerely thank you for your positive recognition of our work's potential impact, which is highly encouraging to us! We also apologize for the suboptimal reading experience in the initial submission. Due to the page limit imposed by the ICLR conference, we had to move some content to the appendix in the original version. We fully agree with your comments and have thoroughly reorganized the manuscript accordingly. **In the newly uploaded version, we have integrated key training details and more ablations into the main paper then streamlined the language throughout**, please feel free to check out our revised manuscript 😊.
>
> > 📝 W1.1: In the current version, even basic information about the training approach, including the losses, learning algorithm, and datasets are left entirely to the appendix. In contrast, the method section focuses almost exclusively on the architecture and has multiple paragraphs devoted to equations ablations the authors will run (e.g. Sec 2.4).
> In my opinion, this section could be reduced to equation 5 plus a little context, and the ablations could be moved to the experiments or appendix. This would make room for information about the training of FALCON.
>
> 💡 **A:** Thank you for this constructive suggestion. We agree that integrating key training details into the main paper is crucial for clarity. As you recommended, we have reorganized the **Method section** in the revised manuscript:
>
> 1. We have added a new subsection **Sec. 2.3 Training Objective**, which introduces the composite loss function **(Eq. 2)** and outlines the two-stage post-training pipeline used by FALCON. This ensures that **the core training methodology is now presented in the main paper**.
> 2. We have significantly streamlined the **Modality Fusion Strategies** discussion in **Sec. 2.5** (previous 2.4). As you suggested, this part has been reduced to **Eq. 6** with core context, confirming *element-wise addition* as our chosen method. The detailed ablation and discussion of alternative fusion mechanisms have been moved to **Sec. 3.3**. We believe the revised organization could greatly improve the reader's experience.

---

> ### Author Response · Authors · 2025-11-24
> **Rebuttal 1: W1-continue**
>
> > 📝 W1.2: Parts of the main paper are especially wordy and flowery. Phrases like "Seamlessly integrating" are very common from LLMs but are hard to evaluate technically -- used three times in the paper. Wording like this could be cut to make space for some of the interesting ablations and details from the supplementary. I actually preferred the more direct writing in the supplementary!
>
> 💡 **A:** We thank you for this critical feedback and apologize for the wordy and flowery language used in our initial submission. Previous use of “seamlessly integrating” was intended to emphasize that the *element-wise addition* effectively integrates spatial information into the VLA. We have performed a thorough revision of the manuscript to eliminate/change such descriptions. Specific revisions made in response to this comment include:
>
> 1. **Targeted modification of "Seamlessly Integrating":**
>     * **L29-30**: "a vision-language-action model that achieves robust 3D spatial understanding by **seamlessly integrating**..." has been revised to "...by **effectively integrating**..."
>     * **L455-456**: "...parameter-free fusion strategy for **seamlessly integrating** spatial and semantic..." has been changed to "...for **combining** spatial and semantic..."
>     * **L755**: "...enabling **seamless integration** of 3D geometric cues." has been replaced with the more factual "enabling **efficient integration** of ..."
> 2. **Another instance of flowery language we revised:** The claim at **L30-32**: "FALCON demonstrates **exceptional** modality transferability by **excelling** with..." was toned down to the more objective "FALCON demonstrates **notable** modality transferability by **performing robustly** with..."
> 3. **More ablations:** We have moved crucial ablation studies from the appendix into the main paper. In **Sec. 3.3**, We have added two new paragraphs: **Spatial Token Injection** and **Modality Fusion Strategies**, which provide the empirical justification for our key design choices in FALCON.

---

> ### Author Response · Authors · 2025-11-24
> **Rebuttal 2: W2.1**
>
> ### **2. Cross-modality Generalization**
> > 📝 W2.1: There are lots of claims and experiments about cross-modality generalization, but I didn't find the experiments especially convincing and I don't feel these claims add much to the paper in their current form. Or maybe the authors could clarify the definition of "modality transferability" in the paper.
> I would like modality transferability experiments showing that a task could be learned (ABCD->D) entirely without depth, and then benefit from depth at test-time (and/or be learned with depth, and not suffer too much from losing depth at test-time) -- these would be more convincing than the current table 5.
>
> 💡 **A:** We sincerely thank you for this valuable feedback and apologize for any confusion caused by our initial presentation of the modality transferability experiments. We have taken substantial steps to address your concerns:
>
> 1. **Clarified definition:** We have provided a clearer definition of "modality transferability" in the revised manuscript. It is now explicitly defined as "**the ability to function and improve under different input modalities (RGB-only, RGB-D, point clouds, camera pose) without retraining or collapsing.**" This definition is introduced and bolded in the **Sec. 1 Introduction (L75-L76)** and reinforced in the **Sec. 3 Experiments**.
> 2. **Further explanation of ESM's role:** We have revised the introduction surrounding the Embodied Spatial Model (ESM) zero-shot depth estimation experiments (now in **Table 6** at **Sec. 3.3**). We clarify that this experiment demonstrates the foundational capability of the ESM module to natively absorb and benefit from diverse geometric inputs (RGB, RGB-D, pose) **within a single, fixed model**. This inherent flexibility of the ESM is the core enabler of FALCON's modality transferability.
> 3. **New test-time depth input/removal experiments:** Following your suggestion, we have conducted new experiments that evaluate the policy's cross-modality generalization. Specifically, we examine two scenarios: **(1) model trained only on RGB but evaluate with additional depth input (test w/ d), and (2) model trained on RGB-D but evaluate without depth (test w/o d).** As shown in **Table C1** below, the results conclusively show that:
>     * **FALCON trained on RGB-only data (w/ rgb) benefits from the introduction of depth at test-time.**
>     * **FALCON trained on RGB-D data (w/ rgb-d) exhibits minimal performance degradation when depth is removed at test-time.**
>
>     For detailed quantitative analysis, please refer to **L479-485** under **Modality Transferability** part in **Sec. 3.3**.
>
> **Table C1:** **Performance comparison of different modality input on CALVIN benchmark.**
> | Method | Task | SR1 | SR2 | SR3 | SR4 | SR5 | Avg. Len. ↑ |
> |--------|------|---|---|---|---|---|-------------|
> | FALCON (w/ rgb) | ABCD→D | **94.0%** | 86.7% | 80.8% | 76.4% | 70.9% | 4.08 |
> | FALCON (test w/ d) | ABCD→D | 93.8% | 86.6% | **81.8%** | 76.6% | **71.3%** | **4.09** |
> | FALCON (w/ rgb-d) | ABCD→D | **94.0%** | **87.0%** | 81.3% | **76.8%** | 70.3% | **4.09** |
> | FALCON (test w/o d) | ABCD→D | 93.7% | 86.4% | 80.9% | **76.8%** | 69.8% | 4.07 |
> |  |  |  |  |  |  |  |  |
> | FALCON (w/ rgb) | ABC→D | 93.7% | 86.9% | 77.9% | 70.3% | 62.2% | 3.91 |
> | FALCON (test w/ d) | ABC→D | 94.2% | **88.4%** | **79.4%** | 70.9% | 61.8% | 3.95 |
> | FALCON (w/ rgb-d) | ABC→D | **94.7%** | 86.7% | 79.1% | **72.4%** | **64.4%** | **3.97** |
> | FALCON (test w/o d) | ABC→D | 94.0% | 87.0% | 78.6% | 71.4% | 63.8% | 3.95 |

---

> ### Author Response · Authors · 2025-11-24
> **Rebuttal 3: Q2, W2.2**
>
> > 📝 Q2: When training and evaluating FALCON, do the authors use the depth and camera poses from the simulator/Realsense + IK? Or is VGGT used at test-time?
>
> 💡 **A:** Thank you for this important question regarding the specific inputs used during training and evaluation. Below, we provide a detailed clarification of the input modalities used across all experiments in the main paper.
>
> 1. **Simulation benchmarks (CALVIN & SimplerEnv):** For all comparative results in **Table 1-4**, **FALCON was trained and evaluated using only RGB input**. This ensures a fair and direct comparison with other VLAs, which are also typically trained and tested on RGB-only data from these benchmarks. The experiments in **Table 5** and the newly added cross-modality tests were specifically designed to validate FALCON's modality transferability. Here, the (w/ rgb) and (w/ rgb-d) models were trained and evaluated with consistent RGB or RGB-D inputs, respectively. The (test w/ d) and (test w/o d) involve a dynamic change of the input depth modality at test-time relative to the fixed modality used during training.
> 2. **Real-World experiments:** For all primary real-world evaluations, FALCON was trained and evaluated using only RGB input from the side camera to ensure a fair comparison with baseline methods. The experiment in **Figure 7** is the specific exception designed to quantify the benefit of additional modality inputs. In this case only, models were trained and evaluated with the combinations of RGB, depth, and camera pose to explicitly measure the performance gain.
> 3. **Role of ESM and VGGT:** **Across all experiments, the spatial information in FALCON is provided by our proposed ESM**. **VGGT [a] was not used** during the training or evaluation of FALCON on any benchmarks reported in our paper.
>
> > 📝 W2.2: However, I wanted to check my understanding about table 5, which the authors say L430-431:
> "ESM achieves performance comparable to VGGT when using only RGB input. Moreover, its performance improves significantly when additional depth or camera pose information is accessible. This demonstrates the inherent strength of FALCON’s modality transferability"
> VGGT is a pointmap estimation model (no actions), and based on the metrics, it looks like this is performance for depth estimation? So the ESM's depth estimation on CALVIN (which the model was trained on), is similar to VGGT (which VGGT wasn't trained on). And then when the GT depth is passed to the ESM, the depth estimation is close to perfect?
>
> 💡 **A:** We sincerely apologize for the lack of clarity in our initial writing, which caused confusion regarding the interpretation of Table 5 (after revision is **Table 6**). Thank you for allowing us to clarify this critical point, and we have revised the manuscript to explain this experiment (also for the evaluation metric) more precisely (see **Embodied Spatial Model** part in **Sec. 3.3**).
>
> 1. **Zero-Shot evaluation & VGGT comparison:** The results in **Table 6** are from a **zero-shot monocular depth estimation experiment** on the CALVIN benchmark. **Crucially, neither our ESM nor the original VGGT model was fine-tuned on the CALVIN dataset.** The comparable performance under RGB-only input (90.91% vs. VGGT's 91.33% for δ<1.25) demonstrates that our ESM retains the strong geometric priors when 3D inputs are absent.
> 2. **Performance with GT inputs:** Your interpretation is accurate regarding the use of ground-truth (GT) inputs. **When GT depth and camera pose are provided to ESM at test-time, the depth estimation quality becomes more precise (e.g., Abs. Rel drops from 8.61 to 0.87)**. This verifies that the ESM's architecture and training strategy successfully enable it to leverage additional spatial signals when they are available. Additionally, other works such as Pow3R [b] and MapAnything [c] have conducted similar experiments and reached the same conclusions.
>
> [a] VGGT: Visual Geometry Grounded Transformer
>
> [b] Pow3R: Empowering Unconstrained 3D Reconstruction with Camera and Scene Priors
>
> [c] MapAnything: Universal Feed-Forward Metric 3D Reconstruction

---

> ### Author Response · Authors · 2025-11-24
> **Rebuttal 4: Q1**
>
> ### **3. Embodied Spatial Model**
> > 📝 Q1: How come the authors add together img + depth tokens (equation 4) and use this strategy, instead of concatenating and either dropping tokens or masking them (like in flow matching/diffusion)?
>
> 💡 **A:** Thank you for this insightful question regarding our design choice in ESM for fusing depth tokens. We apologize for any confusion caused by our initial explanation and are happy to clarify the rationale behind **Eq. 5** (previously Eq. 4).
>
> 1. **Purpose of the Stochastic Formulation:** The primary goal of **Eq. 5** is to implement our stochastic conditioning strategy during ESM training. The Bernoulli random variables $b_{\text{d}}$ and $b_{\text{p}}$ control whether depth and pose are injected in a given training step. Crucially, when $b_{\text{d}}=0$, the depth tokens $\mathbf{T}\_{\text{dpt}}$ are zeroed out, meaning the Spatial Encoder $\mathcal{E}\_{\text{spl}}$ processes only the original image tokens $\mathbf{T}\_{\text{vis}}$ (same process to the camera token). This is fundamental for ensuring the model remains robust and performs effectively when depth/pose inputs are unavailable at test-time.
> 2. **Rationale for Element-wise Addition:** Concatenation would increase the token sequence length input to the subsequent transformer blocks. This would significantly increase computational cost and memory footprint. Besides, Element-wise Addition is a common and validated practice in related works on 3D scene understanding and reconstruction (Pow3R [b] and MapAnything [c]).
>
> [b] Pow3R: Empowering Unconstrained 3D Reconstruction with Camera and Scene Priors
>
> [c] MapAnything: Universal Feed-Forward Metric 3D Reconstruction

---

> > ### Comment · Reviewer_Byfb · 2025-11-27
> > **Thanks for the explanation and revision; 4->6**
> >
> > Thanks to the authors for the thorough explanations to my questions, and the revisions to the paper.
> >
> > Makes sense that the length constraints meant that details get cut, and with the revisions, this addresses most of my major concerns about presentation.
> >
> > I revised my rating from 4->6, and the reason that I stay at 6 that the proposed improvements are structured as a patch on top of existing VLAs.  Current improvements are demonstrated using a relatively small amount of data, and the paper contains plenty of analysis experiments which helps determine effect size at this data/model scale. This approach may scale up, but another paper would need to show that. In short, I feel there the current paper is a definite contribution, most appropriate for a poster.

---

> > > ### Author Response · Authors · 2025-11-27
> > > **Appreciate your constructive feedback and support throughout the review process 😊**
> > >
> > > We are delighted to hear that our revised manuscript addressed most of your concerns about presentation! We sincerely appreciate your constructive feedback and support throughout the review process. If you have any further questions or suggestions, please feel free to let us know.

---

### Official Review · Reviewer_x5su · 2025-11-01

**Soundness:** 3
**Presentation:** 3
**Contribution:** 2
**Rating:** 6
**Confidence:** 4

**Summary:**

The paper introduces FALCON, a spatial to action paradigm that injects richer and more representative 3D spatial tokens into VLAs through an improved injection scheme. The approach combines a 2D VLM backbone, an embodied spatial model for 3D encoding, and a spatial-enhanced action head for fusion. Experiments on CALVIN, SimplerEnv, and real-world tasks show better performance and improved modality transferability.

**Strengths:**

1. This paper focus on spatial reasoning in VLAs, which is a crucial research problem.
2. The experiments cover multiple benchmarks on both simulation and real-world.
3. Clear qualitative visualizations are provided to support the contribution.
4. The pipeline is complete, which combines 2D VLM backbone, embodied spatial model for 3D encoding, and spatial-enhanced action head for fusion.

**Weaknesses:**

1. The core concept of this paper “3D spatial tokens” is ill-defined. This paper claims that 3D spatial tokens provide robust geometric priors. However, in line 132, the depth information is optional, and in line 224, the depth and/or pose is randomly injected. The resulting spatial tokens Tspl derives from DINO and an encoder with cross/self-attention. Thus, these spatial tokens are not truly 3D, they are 2D correlations.
2. In Eq. 4, they randomly inject depth and/or pose. The proposed stochastic conditioning breaks the well-defined mapping of Eq. 1. So, the expectations or joint distribution forms of F should be redefined.
3. The training pipeline requires further explanation. In appendix A.3, Stage 1 freezes all pre-trained components and optimize only the adapter parameters ΘD. The input for D is fixed (as ESM output is frozen), this training does not really learn cross-space alignment, it merely compresses the ESM's output. Stage 2 unfreezes VLM parameters ΘV and adapter parameters ΘD, while keeps ΘA and ΘG frozen. If my understanding is correct, here although VLM can learn new representations, the action control component is not adapted to new spatial features. So, the model cannot learn how spatial features influence actions, thus the Loss L cannot truly drive the entire system to learn the mapping from semantics and spatial information to action.

**Questions:**

1. As weakness 3, can you explain why freeze ΘA and ΘG in stage 2?
2. In Eq. 6, λ balances MSE and BCE losses, but its selection criteria are not described. Is λ chosen empirically, through grid search, or via adaptive re-weighting? How sensitive is the model’s convergence and performance to λ?
3. The ESM uses multi-task supervision, how each supervision type contributes to downstream VLA performance? What happens if one component is removed?
4. The pipeline adopts a two-stage post-training approach rather than joint training, how about joint end-to-end optimization? Why do two-stage?
5. Is the sampling probability p in Eq. 4 fixed over training?

---

> ### Author Response · Authors · 2025-11-24
> **Rebuttal 1: W1, W2**
>
> Thank you for your thoughtful review and we appreciate your acknowledgment that our paper “focuses on a crucial research problem,” offers “clear qualitative visualizations,” and presents a “complete pipeline.” These encouraging remarks motivate us to further improve the manuscript. In the revised manuscript, we mark our major revisions as $\textbf{\textcolor{blue}{blue}}$ for your easy reference. Additionally, the weaknesses and questions you raised are insightful, and we have carefully addressed them below.
>
> > 📝 W1: The core concept of this paper “3D spatial tokens” is ill-defined. This paper claims that 3D spatial tokens provide robust geometric priors. However, in line 132, the depth information is optional, and in line 224, the depth and/or pose is randomly injected. The resulting spatial tokens Tspl derives from DINO and an encoder with cross/self-attention. Thus, these spatial tokens are not truly 3D, they are 2D correlations.
>
> 💡 **A:** We sincerely thank you for this critical feedback and apologize for any confusion caused by our initial presentation of “3D spatial tokens”. We agree that the term “3D spatial tokens” deserves clarification. The spatial tokens derived from Embodied Spatial Model (ESM) do not represent explicit 3D coordinates, but rather encode rich geometric priors that enable the model to reason about 3D structure. This is consistent with recent spatial foundation models (e.g., VGGT [a], DUSt3R [b], and MASt3R [c]) that reconstruct 3D scenes from 2D images using implicit geometric cues. Specifically:
>
> * The tokens are derived from a spatial encoder trained with multi-task 3D supervision (depth estimation, point maps prediction, camera pose prediction), enabling them to capture geometry-aware representations.
> * When depth or pose is available, these tokens are enhanced via stochastic conditioning (**Eq. 5** in the revised paper), allowing the model to leverage stronger geometric signals without being dependent on them.
>
> Thus, while the tokens are not “3D” in the explicit sense, they encode 3D-aware priors that are helpful for spatial perception in robotic tasks.
>
> > 📝 W2: In Eq. 4, they randomly inject depth and/or pose. The proposed stochastic conditioning breaks the well-defined mapping of Eq. 1. So, the expectations or joint distribution forms of F should be redefined.
>
> 💡 **A:** We apologize for the errors in our original formulation of **Eq. 1** and sincerely thank you for pointing this out. We have fixed **Eq. 1** in the revised manuscript by explicitly denoting the depth map $D_t$ and camera pose $P$ are optional inputs. The revised **Eq. 1** is now presented as:
>
> $$
> A_t = \mathcal{F}(O_t, L, D_t, P), \quad \text{where } D_t \text{ and } P \text{ are optional}.
> $$
>
> [a] VGGT: Visual Geometry Grounded Transformer
>
> [b] DUSt3R: Geometric 3D Vision Made Easy
>
> [c] Grounding Image Matching in 3D with MASt3R

---

> ### Author Response · Authors · 2025-11-24
> **Rebuttal 2: W3.1, Q4**
>
> > 📝 W3.1: The training pipeline requires further explanation. In appendix A.3, Stage 1 freezes all pre-trained components and optimize only the adapter parameters ΘD. The input for D is fixed (as ESM output is frozen), this training does not really learn cross-space alignment, it merely compresses the ESM's output.
>
> 💡 **A:** We sincerely apologize for the lack of clarity in explaining our **multi-stage training pipeline**. Your feedback is insightful, and we appreciate the opportunity to clarify.
>
> Let's focus on **Stage 1** in this part. The design of **Stage 1** is inspired by the training paradigms in works like LLaVA [d] and PointVLA [e], **where only a lightweight projection module is trained to align a new modality (e.g., visual or spatial tokens) with a frozen pre-trained model (e.g., LLM or VLA).** Specifically:
>
> 1. In **Stage 1**, we freeze all pre-trained components (VLM $\mathcal{V}$, action predictor $\pi$, and ESM $\mathcal{G}$) and **optimize only the adapter $\mathcal{D}$ parameters $\Theta_D$**, which is a deliberate strategy to align the spatial tokens with the feature space of the VLA without disrupting pre-trained representations. Besides, $\mathcal{D}$ employs a **zero-initialized final linear layer**, which enabling **gradual integration** of the geometric features and ensuring stable optimization.
> 2. This approach is analogous to LLaVA’s **training Stage 1**, where a projection layer aligns **image features derived from a frozen visual encoder** with a **frozen LLM’s word embeddings**. Similarly, in PointVLA, lightweight injectors are trained to fuse 3D features **without retraining the VLA backbone**.
> 3. Empirically, this alignment is effective: as shown in **Table B1**, after **Stage 1**, FALCON’s performance improves (**Avg. Len. increases from 3.48 to 3.66**), demonstrating that **Stage 1 can be understood as training a compatible spatial encoder for the frozen VLA**, it serves as a stable and efficient initialization for subsequent joint refinement in **Stage 2**.
>
> **Table B1:** **Performance comparison of pre-trained VLA and FALCON Stage 1 on CALVIN benchmark.**
> | Method | Task | SR1 | SR2 | SR3 | SR4 | SR5 | Avg. Len. ↑ |
> |--------|------|---|---|---|---|---|-------------|
> | Kosmos-VLA (w/ rgb) | ABC→D | **91.6%** | 80.0% | 68.3% | 58.8% | 49.0% | 3.48 |
> | **FALCON (Stage 1)** | ABC→D | 91.5% | **81.8%** | **72.4%** | **64.6%** | **55.6%** | **3.66** |
>
> > 📝 Q4: The pipeline adopts a two-stage post-training approach rather than joint training, how about joint end-to-end optimization? Why do two-stage?
>
> 💡 **A:** Thank you for this important question. We adopted a two-stage training approach primarily to **ensure stable and effective integration of the new spatial features** without disrupting the pre-trained model's representations as we mentioned in the answer to **W3.1**.
>
> Besides, as **the adapter $\mathcal{D}$ is randomly initialized**, which would introduce **noisy gradients** if trained jointly end-to-end, and this could interfere with both the fine-tuning of the VLA and the effective incorporation of geometric conditions, leading to suboptimal performance.
>
> Empirically, as shown in **Table B2**, direct joint end-to-end optimization from the same pre-trained checkpoint (*joint training*) results in an Avg. Len. of 3.78, which is lower than our two-stage pipeline. In contrast, our method **first warming up $\mathcal{D}$ in Stage 1 (Avg. Len. improves from 3.48 to 3.66)**, and then **further refines the $\Theta_V$ and $\Theta_D$ in** **Stage 2** (**Avg. Len. reaches 3.91**), which demonstrates that **the two-stage strategy progressively enhances model performance**.
>
> **Table B2:** **Performance comparison of different training paradigms and stages on CALVIN benchmark.**
> | Method | Task | SR1 | SR2 | SR3 | SR4 | SR5 | Avg. Len. ↑ |
> |--------|------|---|---|---|---|---|-------------|
> | Kosmos-VLA (w/ rgb) | ABC→D | 91.6% | 80.0% | 68.3% | 58.8% | 49.0% | 3.48 |
> | FALCON (Stage 1) | ABC→D | 91.5% | 81.8% | 72.4% | 64.6% | 55.6% | 3.66 |
> | FALCON (joint training) | ABC→D | 93.4% | 83.1% | 74.2% | 67.2% | 60.2% | 3.78 |
> | **FALCON (ours)** | ABC→D | **93.7%** | **86.9%** | **77.9%** | **70.3%** | **62.2%** | **3.91** |
>
> [d] Visual Instruction Tuning
>
> [e] PointVLA: Injecting the 3D World into Vision-Language-Action Models

---

> ### Author Response · Authors · 2025-11-24
> **Rebuttal 3: W3.2, Q1**
>
> > 📝 W3.2: Stage 2 unfreezes VLM parameters ΘV and adapter parameters ΘD, while keeps ΘA and ΘG frozen. If my understanding is correct, here although VLM can learn new representations, the action control component is not adapted to new spatial features. So, the model cannot learn how spatial features influence actions, thus the Loss L cannot truly drive the entire system to learn the mapping from semantics and spatial information to action.
>
> > 📝 Q1: As weakness 3, can you explain why freeze ΘA and ΘG in stage 2?
>
> 💡 **A:** Thank you for raising these crital feedback and questions regarding why freezing action predictor $\pi$ and ESM $\mathcal{G}$'s parameters $\Theta_A$ and $\Theta_G$ in **Stage 2**. We apologize for any confusion and are happy to clarify the rationale behind this design.
>
> 1. Firstly, the decision to freeze $\Theta_A$ and $\Theta_G$ in **Stage 2** draws inspiration from established practices in modular VLA training, such as PointVLA [e] and DexVLA [f]. The key insight is that **both components are pre-trained on large-scale datasets and already possess strong capabilities**: $\Theta_A$ in **mapping semantic features to actions**, $\Theta_G$ in **providing robust geometric priors**. Fine-tuning them on limited data could lead to overfitting or catastrophic forgetting of their generalizable knowledge. Instead, by freezing $\Theta_A$ and $\Theta_G$, we allow the VLM ($\Theta_V$) and adapter ($\Theta_D$) to further adapt to the aligned spatial-semantic features from **Stage 1**, enabling the model to learn how spatial information influences actions indirectly through the refined semantic representations. This ensures efficient optimization and preserves the integrity of pre-trained components.
> 2. Empirically, this approach is justified by our ablation studies in **Table B3**. As shown below, our full method achieves **the highest performance (SR5: 62.2%, Avg. Len.: 3.91)**, outperforming the following variants: (1) **(Stage 2-head) tune $\Theta_A$ and $\Theta_D$ while freeze $\Theta_V$ and $\Theta_G$ in Stage 2**, (2) **(Stage 2-all) unfreeze all components except $\Theta_G$ in Stage 2**. This demonstrates that our standard paradigm offers the optimal solution.
> 3. Moreover, compared to from the same pre-trained checkpoint, **continuing tune only the VLM ($\Theta_V$) from Kosmos-VLA (w/ rgb, w/o ESM) for the same epochs**, our method achieves a significant improvement **(Avg. Len. 3.45 vs. 3.91)**. This demonstrates that spatial features are effectively integrated to enhance action generation, as the **VLM implicitly refines it's semantic features to incorporate spatial cues, which subsequently guide the action predictor**. Thus, our design strategically balances performance and computational efficiency by preserving pre-trained components while enabling spatial information to influence actions.
>
> **Table B3:** **Performance comparison of different training paradigms on CALVIN benchmark.**
> | Method | Task | SR1 | SR2 | SR3 | SR4 | SR5 | Avg. Len. ↑ |
> |--------|------|---|---|---|---|---|-------------|
> | Kosmos-VLA (tune VLM) | ABC→D | 90.7% | 78.4% | 67.7% | 58.4% | 49.8% | 3.45 |
> | FALCON (Stage 2-head) | ABC→D | 92.3% | 83.2% | 72.8% | 63.1% | 54.5% | 3.66 |
> | FALCON (Stage 2-all) | ABC→D | 93.3% | 84.7% | 75.2% | 68.0% | 60.5% | 3.81 |
> | **FALCON (ours)** | ABC→D | **93.7%** | **86.9%** | **77.9%** | **70.3%** | **62.2%** | **3.91** |
>
> [e] PointVLA: Injecting the 3D World into Vision-Language-Action Models
>
> [f] DexVLA: Vision-Language Model with Plug-In Diffusion Expert for General Robot Control

---

> ### Author Response · Authors · 2025-11-24
> **Rebuttal 4: Q2, Q3, Q5**
>
> > 📝 Q2: In Eq. 6, λ balances MSE and BCE losses, but its selection criteria are not described. Is λ chosen empirically, through grid search, or via adaptive re-weighting? How sensitive is the model’s convergence and performance to λ?
>
> 💡 **A:** Thanks for your careful review and pointing out the omission regarding the selection of λ in **Eq. 2** (previous Eq. 6). We apologize for not providing sufficient details on this hyperparameter and appreciate for giving us the opportunity to provide further explanation.
>
> 1. For all experiments in our paper, we **following the settings in RoboVLM [g] to set λ = 0.01** for fair comparisons. We have also added a clarification in **training details of FALCON at Appendix A.4 (L812-813)** of the revised manuscript to explain this choice.
> 2. To further address your concern about sensitivity of different λ, we conducted an additional experiment for FALCON with **λ = 0.05** under RGB-only input on CALVIN zero-shot setting. As shown in **Table B4**, performance decreased slightly when λ increased from 0.01 to 0.05 **(Avg. Len. dropped from 3.91 to 3.87)**. Analysis of the overall train loss curves (**Figure 10 in the Appendix A.7.1**) reveals that **larger λ leads to increased oscillation during training**, while the overall convergence trend remains the same, which indicates that **the model is not highly sensitive to small variations in λ**.
>
> **Table B4:** **Performance comparison of different loss weighting factors on CALVIN benchmark.**
> | Method | Task | SR1 | SR2 | SR3 | SR4 | SR5 | Avg. Len. ↑ |
> |--------|------|---|---|---|---|---|-------------|
> | FALCON (λ = 0.05) | ABC→D | 93.2% | 85.8% | 77.4% | 69.5% | 60.4% | 3.87 |
> | **FALCON (λ = 0.01)** | ABC→D | **93.7%** | **86.9%** | **77.9%** | **70.3%** | **62.2%** | **3.91** |
>
> > 📝 Q3: The ESM uses multi-task supervision, how each supervision type contributes to downstream VLA performance? What happens if one component is removed?
>
> 💡 **A:** Thank you for this insightful question. Our ESM is finetuned on VGGT, which has already been trained with multi-task supervision. Besides, the ESM remains frozen during VLA training. Therefore, even if we perform ablations on the supervision tasks during the ESM finetuning stage, the results would still be influenced by the original VGGT checkpoint, as these geometric representations are already inherently fused during its pre-training.
>
> > 📝 Q5: Is the sampling probability p in Eq. 4 fixed over training?
>
> 💡 **A:** Thank you for this important question and we apologize for not explicitly stating the value of the sampling probability $p$ in the original manuscript. To clarify, the $p$ in $b_{\text{d}}, b_{\text{p}} \sim \text{Bernoulli}(p)$ **is fixed throughout ESM training**. We set **$p = 66\\%$** to ensure the model is frequently exposed to geometric conditions while still maintaining robustness to RGB-only inputs. We have now added this clarification in **Appendix A.3.2 (L790)** of the revised paper to prevent any ambiguity.
>
> [g] Towards Generalist Robot Policies: What Matters in Building Vision-Language-Action Models

---

### Official Review · Reviewer_xvie · 2025-11-01

**Soundness:** 3
**Presentation:** 3
**Contribution:** 3
**Rating:** 8
**Confidence:** 4

**Summary:**

The paper proposes a new VLA model, FALCON, which tries to incorporate the spatial 3D information into the VLA implicitly. The goal is to be able to utilize 3D information, if available, or otherwise be robust to different modality setups (full 3D, depth only, camera only, no 3D). The idea to achieve this is to utilize 3D feature information in models like VGGT, and additionally supply 3D information to the encoders of these models if they are available. Then, these spatial features are added to the action features before passing to the action head. This lets the action head utilize the 3D information for predicting actions. The paper conducts experiments on CALVIN, SIMPLER and real-world tasks. They show that the proposed method works better than prior 3D policies and 2D/3D VLAs, and the ablations show that the model works well in various modality setups and can leverage 3D information to improve performance, when available.

**Strengths:**

- The paper is very well-written and easy to follow
- The model design and ideas are clearly novel and overall make sense
- The problem setting is interesting and important — to utilize 3D information when available, but be robust in cases when it is not available.
- The ablations are adequate and clearly show that the model is robust to various modality scenarios and can benefit form 3D data, when available.

**Weaknesses:**

My main concern is that the evaluation benchmarks chosen, especially the simulation ones are weak and do not necessarily benefit from 3D understanding and information.

For instance, the paper makes this statement: “FALCON surpasses previous methods that rely on ground-truth point clouds (e.g., 3DDP (Ze et al., 2024) and 3D Diffuser Actor (Ke et al., 2024)), improving the Avg.
Len. by 4.13 and 1.05, respectively. This provides clear evidence of the effectiveness of our implicit
spatial information integration strategy” This statement appears wrong because the ablations in table-4 show that actually on Calvin, the performance is similar between RGB-only or RGB-D versions of their model, showing that additional 3D input doesn’t help much in this benchmark. And while it can be argued that FALCON’s implicit geometry is enough and that’s why explicit 3D doesn’t help, that is probably not true as the same ablations show that in real-world benchmarks performance increase from 60 to 80%. This makes me think that it is just that Calvin tasks don’t benefit much from 3D. Thus, it is unclear if the gains over prior methods especially the explicit 3D models like 3DDA is due to the implicit spatial information strategy or more parameters / better data. Concretely speaking, it is not convincing if implicit 3D, as proposed by this method, is enough or we need more richer 3D-aware models.

It would be more convincing to show results on RL-Bench which contain several high precision tasks and where 3D policies (like 3DDA) have worked significantly better than 2D policies. Experiments there would help understand if the implicit 3D is strong enough or not.


(Minor)
Table-5 looks like it is testing monocular depth estimation and camera pose estimation — however that is not obvious from the description. It might help to explicitly say this

**Questions:**

As I describe in the weakness section, I suggest testing the proposed methods on richer settings like RL-Bench and compare with explicit 3D methods to eke out if the proposed implicit 3D representation is enough.

---

> ### Author Response · Authors · 2025-11-24
> **Rebuttal 1: W1**
>
> We sincerely thank you for your thorough and constructive feedback, as well as your positive assessment of the novelty, clarity, and experimental rigor of our work. Below, we provide a point-by-point response to the main concerns you raised and we mark our major revisions as $\textbf{\textcolor{blue}{blue}}$ in the updated manuscript for your easy reference.
>
> > 📝 W1: My main concern is that the evaluation benchmarks chosen, especially the simulation ones are weak and do not necessarily benefit from 3D understanding and information.
> For instance, the paper makes this statement: “FALCON surpasses previous methods that rely on ground-truth point clouds (e.g., 3DDP (Ze et al., 2024) and 3D Diffuser Actor (Ke et al., 2024)), improving the Avg. Len. by 4.13 and 1.05, respectively. This provides clear evidence of the effectiveness of our implicit spatial information integration strategy” This statement appears wrong because the ablations in table-4 show that actually on Calvin, the performance is similar between RGB-only or RGB-D versions of their model, showing that additional 3D input doesn’t help much in this benchmark. And while it can be argued that FALCON’s implicit geometry is enough and that’s why explicit 3D doesn’t help, that is probably not true as the same ablations show that in real-world benchmarks performance increase from 60 to 80%. This makes me think that it is just that Calvin tasks don’t benefit much from 3D. Thus, it is unclear if the gains over prior methods especially the explicit 3D models like 3DDA is due to the implicit spatial information strategy or more parameters / better data. Concretely speaking, it is not convincing if implicit 3D, as proposed by this method, is enough or we need more richer 3D-aware models.
>
> 💡 **A:** We sincerely thank you for raising this important point regarding the role of 3D information in the CALVIN benchmark and the interpretation of our results. We agree that a deeper investigation is necessary to clarify these aspects. To address your concern, we have conducted additional experiments on CALVIN, and our findings demonstrate the value of spatial information in this benchmark.
>
> 1. **CALVIN does benefit from spatial information:** Your observation that FALCON performs comparable under RGB and RGB-D (point clouds) settings in CALVIN is accurate, and this actually highlights the strength of our implicit spatial strategy, which **achieves robust performance even when geometric inputs are absent**. To further verify whether CALVIN tasks can benefit from spatial information, we performed a controlled comparison under identical settings using **Kosmos-VLA (w/ rgb)**, which is a 2D VLA baseline. As shown in **Table A1**, introducing explicit 3D inputs (point clouds) to Kosmos-VLA leads to clear improvements, **especially in the challenging zero-shot setting ABC→D**: **Success Rate at task 5 (SR5) increases from 49.0% to 66.3%, and Avg. Len. improves from 3.48 to 3.98**. This confirms that CALVIN tasks do benefit from 3D information when the model can effectively leverage it.
>
> 2. **Modality transferability and robustness**: The primary advantages of FALCON is not significantly outperforming explicit 3D models under the same conditions (model parameters, data), but rather **delivering competitive performance using only RGB, while retaining the flexibility to benefit from additional 3D inputs when available, all without retraining**. To further demonstrate this, we evaluated FALCON under varying test-time depth conditions while keeping the training modality fixed. As summarized in **Table A1**:
>     * **When trained on RGB only but tested with additinal depth inputs (test w/ d), performance improves (e.g., ABC→D: Avg. Len. increased from 3.91 to 3.95)**, matching the performance of model trained with full RGB-D (w/ rgb-d).
>     * **When trained on RGB-D but tested without depth (test w/o d), performance remains highly robust.**
>
>     This shows that FALCON can **dynamically leverage available geometric signals at test-time**, achieving performance comparable to models trained explicitly with RGB-D/point clouds, while maintaining robustness across modality changes.

---

> > ### Author Response · Authors · 2025-11-24
> > **Rebuttal 1: W1-continue**
> >
> > **Table A1:** **Performance comparison of different modality input on CALVIN benchmark.**
> > | Method | Task | SR1 | SR2 | SR3 | SR4 | SR5 | Avg. Len. ↑ |
> > |--------|------|---|---|---|---|---|-------------|
> > | Kosmos-VLA (w/ rgb) | ABCD→D | 92.5% | 85.4% | 79.1% | 74.9% | 68.7% | 4.01 |
> > | Kosmos-VLA (w/ rgb-d) | ABCD→D | 92.4% | 85.3% | 80.0% | 76.5% | 70.5% | 4.05 |
> > | FALCON (w/ rgb) | ABCD→D | **94.0%** | 86.7% | 80.8% | 76.4% | 70.9% | 4.08 |
> > | FALCON (test w/ d) | ABCD→D | 93.8% | 86.6% | **81.8%** | 76.6% | **71.3%** | **4.09** |
> > | FALCON (w/ rgb-d) | ABCD→D | **94.0%** | **87.0%** | 81.3% | **76.8%** | 70.3% | **4.09** |
> > | FALCON (test w/o d) | ABCD→D | 93.7% | 86.4% | 80.9% | **76.8%** | 69.8% | 4.07 |
> > |  |  |  |  |  |  |  |  |
> > | Kosmos-VLA (w/ rgb) | ABC→D | 91.6% | 80.0% | 68.3% | 58.8% | 49.0% | 3.48 |
> > | Kosmos-VLA (w/ rgb-d) | ABC→D | 93.6% | 86.0% | 78.6% | **73.3%** | **66.3%** | **3.98** |
> > | FALCON (w/ rgb) | ABC→D | 93.7% | 86.9% | 77.9% | 70.3% | 62.2% | 3.91 |
> > | FALCON (test w/ d) | ABC→D | 94.2% | **88.4%** | **79.4%** | 70.9% | 61.8% | 3.95 |
> > | FALCON (w/ rgb-d) | ABC→D | **94.7%** | 86.7% | 79.1% | 72.4% | 64.4% | 3.97 |
> > | FALCON (test w/o d) | ABC→D | 94.0% | 87.0% | 78.6% | 71.4% | 63.8% | 3.95 |

---

> ### Author Response · Authors · 2025-11-24
> **Rebuttal 2: W2**
>
> > 📝 W2: (Minor) Table-5 looks like it is testing monocular depth estimation and camera pose estimation — however that is not obvious from the description. It might help to explicitly say this.
>
> 💡 **A:** We sincerely thank you for this valuable feedback and apologize for the oversight that the original description of Table 5 (now **Table 6** in the revised manuscript) lacked clarity regarding its evaluation tasks and metrics. We have enriched the caption of **Table 6** and added description to introduce this experiment in **Sec. 3.3 (L501-502)**. Specifically, **Table 6** evaluates the **zero-shot monocular depth estimation capability of Embodied Spatial Model (ESM) on CALVIN benchmark**. The results demonstrate that our ESM **achieves competitive performance using only RGB input, and shows significant improvement when additional depth and camera pose are available**. This quantitatively validates **the effectiveness of the implicit spatial features derived from ESM**.

---

### Author Response · Authors · 2025-12-01
**General Response**

We sincerely thank all reviewers, AC, SAC, and PC for their precious time, thoughtful feedback, and engagement throughout the review period.

Firstly, we would like to once again highlight the key motivation and novelty of this paper. We hope this helps in better understanding of our work:

**Motivation and Novelty.** Existing VLA models act in 3D real-world but are typically built on 2D encoders, leaving a spatial reasoning gap that limits generalization and adaptability. Recent 3D integration techniques for VLAs either **require specialized sensors and transfer poorly across modalities**, or **inject weak cues that lack geometry and degrade vision-language alignment**. In this work, we introduce **FALCON (From Spatial to Action)**, which overcomes these issues by **decoupling spatial processing from the VLM**, preserving its semantic integrity while presenting notable spatial understanding. Compared to existing baselines, FALCON achieves **strong modality transferability** and robust performance across diverse sensory conditions, **reduces dependency on specialized sensor setups** when deploy in the real-world.

Secondly, we have submitted a revised manuscript that fully incorporates the reviewers’ feedback and includes additional experiments conducted during the rebuttal. All revised sections in the manuscript are highlighted in $\textbf{\textcolor{blue}{blue}}$ for easy reference. In summary:

**1. Highlighted empirical validation and discussion:**
* Extended modality transferability experiments on CALVIN with varying test-time depth conditions in **Sec. 3.3 and Table 5** (Reviewer Byfb).
* New 2D VLA baseline evaluations on CALVIN to validate the importance of spatial information in **Sec. 3.3 and Table 5** (Reviewer xvie).
* New experiments on CALVIN to validate the selection of key hyperparameters in FALCON: (a) loss weighting factors in **Appendix A.7.1**, (b) Bernoulli probabilities in **Appendix A.7.5** (Reviewer x5su and jhgj).
* New ablations on CALVIN to investigate the effect of the LSTM history length H and chunk size C on stability and task success rate, which are presented in **Appendix A.7.2** (Reviewer jhgj).
* New ablations on CALVIN under different training paradims for validating our multi-stage training pipeline (Reviewer x5su).

**2. Improved paper organization and methodology clarity:**
* Integrated key training details in **Sec. 2.3** regarding the composite loss function (**Eq. 2**) and two-stage post-training pipeline used by FALCON, shorten the discussion of different Modality Fusion Strategies in **Sec. 2.5** (Reviewer Byfb).
* Incorporated ablations providing empirical justification for our key design choices in FALCON into the main paper (**Sec. 3.3**) then streamlined the language throughout (Reviewer Byfb).

**3. Additional real-world validation:**
* Extended data efficiency experiments on two real-world few-shot tasks with varying number of training demonstrations in **Appendix A.7.3** (Reviewer jhgj).

**Summary of the reviewer discussion period:**

Finally, following our item-to-item responses, **Reviewer Byfb indicated that the major concerns about presentation had been fully addressed and raised the score (Rating $4 \rightarrow 6$), other three reviewers (Ratings 6, 8, 8) have not yet responded.** We believe the revisions and additional experiments outlined above effectively address the reviewers’ concerns and strengthen the paper. We hope these changes further clarify our contributions and the robustness of our findings.

---

### Meta-Review · Area_Chair_gPHA · 2026-01-03

**Summary:**

The reviewers recognized the importance of the problem addressed: bridging the 2D-3D spatial reasoning gap in Vision-Language-Action (VLA) models. The primary concerns that informed the discussion are:
- Initial confusion regarding the definition of "3D spatial tokens" and errors in the mathematical formulation of the stochastic conditioning.
- Skepticism regarding whether current simulation benchmarks (like CALVIN) truly benefit from 3D information or if the model’s gains were due to parameter scaling.
- The need for empirical and theoretical grounding for specific design choices, such as the two-stage post-training pipeline and the use of element-wise addition for modality fusion.
- Concerns about "flowery" language and the need for a clearer comparison with contemporaneous 3D-enhanced VLAs (e.g., GeoVLA, PointVLA).

**Reviewer Concerns:**

Concerns Addressed by Rebuttal:
- Terminology and Math: The authors clarified that "3D tokens" represent implicit geometric priors from spatial foundation models. They corrected the mathematical mapping in equations, which now explicitly accounts for optional depth/pose inputs.
- Presentation: Reviewer Byfb’s concerns regarding organization and wordiness were directly addressed. The corresponding sections were reorganized to bring training details from the appendix to the main paper, and "LLM-style" adjectives were removed.
- Fusion Strategy: In response to Reviewer jhgj, the authors provided a sophisticated "alignment drift" analysis (Table D2), demonstrating that element-wise addition preserves the VLM's semantic space better than cross-attention (lowest L2 distance: 10.05).
- Modality Transferability: New experiments (Table C1) convincingly showed that FALCON can benefit from depth at test-time even when trained on RGB-only data, and remains robust when depth is removed.
- Two-Stage Training: Authors provided new ablations (Table B1/B2) justifying the two-stage approach as a means to ensure stable integration without disrupting pre-trained VLM representations.

Outstanding Concerns:
- Simulation Benchmarks: Reviewer xvie suggested testing on RL-Bench for richer 3D precision tasks. While the authors provided new evidence on CALVIN (Table A1) and 11 real-world tasks to show 3D utility, RL-Bench results were not included in the revision. However, the comprehensive real-world evaluation significantly mitigates this concern.

**Reviewer Scores:**

- Reviewer xvie (Score 8): Likely would have maintained an 8. The reviewer was already positive, and the authors provided the requested analysis.
- Reviewer jhgj (Score 8): Likely would have maintained an 8. The reviewer’s high-level technical questions were met with high-quality new data.
- Reviewer Byfb (Score raised from 4 to 6): This reviewer already officially raised their score after the rebuttal, citing that the revisions addressed most major concerns about presentation and experimental clarity.
- Reviewer x5su (Score 6): Likely would have maintained a 6 or even raised their score. Most of this reviewer’s "Weaknesses" were technical/clarity-based. The authors provided a very thorough response and corrected the formal errors the reviewer identified.

---

### Decision · Program_Chairs · 2026-01-26

Accept (Poster)